# A conserved N-terminal motif of CUL3 contributes to assembly and E3 ligase activity of CRL3$^{KLHL22}$

Weize Wang [1,2,6], Ling Liang [3,6] ✉, Zonglin Dai [4,6], Peng Zuo [4], Shang Yu[3], Yishuo Lu [2], Dian Ding [4], Hongyi Chen[2], Hui Shan[1], Yan Jin[4], Youdong Mao [2,5] & Yuxin Yin [1,2,4] ✉

The CUL3-RING E3 ubiquitin ligases (CRL3s) play an essential role in response to extracellular nutrition and stress stimuli. The ubiquitin ligase function of CRL3s is activated through dimerization. However, how and why such a dimeric assembly is required for its ligase activity remains elusive. Here, we report the cryo-EM structure of the dimeric CRL3$^{KLHL22}$ complex and reveal a conserved N-terminal motif in CUL3 that contributes to the dimerization assembly and the E3 ligase activity of CRL3$^{KLHL22}$. We show that deletion of the CUL3 N-terminal motif impairs dimeric assembly and the E3 ligase activity of both CRL3$^{KLHL22}$ and several other CRL3s. In addition, we found that the dynamics of dimeric assembly of CRL3$^{KLHL22}$ generates a variable ubiquitination zone, potentially facilitating substrate recognition and ubiquitination. These findings demonstrate that a CUL3 N-terminal motif participates in the assembly process and provide insights into the assembly and activation of CRL3s.

E3 ubiquitin ligases (E3s) play an essential role in maintaining cellular protein homeostasis. It is estimated that there are more than 600 E3s in human cells[1]. Among these, Cullin-RING E3 ubiquitin ligases (CRLs) stand out as multi-subunit complexes comprising the Cullin scaffold subunit, an adapter subunit, a substrate receptor subunit, as well as an E2-recruiting subunit RBX1/2[2,3]. As the biggest family of E3 ubiquitin ligase, CRLs participate in many cellular processes, including cell cycle progression, DNA damage repair, and maintenance of chromatin integrity[4–7]. Dysregulation of CRLs has been associated with various diseases, including neural disorders, hypertension, developmental abnormalities, viral infections, and cancer[8]. Furthermore, the CRL-based targeted protein degradation design, such as the Proteolysis-Targeting Chimera (PROTAC), has shown exciting potential for therapeutic applications in recent years[9]. Thus, structural insights into the

assembly and activation of CRLs may provide important insights into how they may be best exploited for the treatment of CRL-related diseases.

Unlike other CRL families, CUL3-based CRLs (CRL3s) use a single polypeptide function both as an adapter and a substrate receptor, known as BTB (Bric-a-brac, Tramtrack, and Broad complex) domain-containing protein[10–13]. It is estimated that about 183 human BTB domain-containing proteins participate in various cellular processes[14]. Kelch-like ECH-associated protein 1 (KEAP1, also known as KLHL19) is one of the most studied BTB domain-containing proteins. CRL3$^{KEAP1}$ recognizes nuclear factor erythroid 2-related factor 2 (NRF2) and promotes NRF2 polyubiquitination and degradation[15–18]. Under oxidative stress conditions, disulfide bond formation within KEAP1 inhibits the function of CRL3$^{KEAP1}$, which results in NRF2 accumulation. Elevated

[1]Institute of Precision Medicine, Peking University Shenzhen Hospital, Shenzhen 518036, China. [2]Peking-Tsinghua Center for Life Sciences, Peking University, 100871 Beijing, China. [3]Department of Biophysics, School of Basic Medical Sciences, Peking University Health Science Center, 100191 Beijing, China. [4]Institute of Systems Biomedicine, Department of Pathology, Beijing Key Laboratory of Tumor Systems Biology, School of Basic Medical Sciences, Peking University Health Science Center, 100191 Beijing, China. [5]State Key Laboratory for Artificial Microstructures and Mesoscopic Physics, Center for Quantitative Biology, National Biomedical Imaging Center, School of Physics, Peking University, 100871 Beijing, China. [6]These authors contributed equally: Weize Wang, Ling Liang, Zonglin Dai. ✉e-mail: liangling@bjmu.edu.cn; yinyuxin@hsc.pku.edu.cn

NRF2 protein levels promote transcription of oxidase-resistant genes and protect cells from oxidative stress. In contrast, the BTB-containing Kelch-like protein 22 (KLHL22) plays important roles in mitosis, T-cell activity and amino acid sensing[19–21]. CRL3[KLHL22] ubiquitylates several cellular substrates, including serine/threonine-protein kinase PLK1[19], programmed cell death protein 1 (PD1)[20], and DEP domain-containing protein 5 (DEPDC5)[21]. DEPDC5 is the largest subunit of the GAP activity toward Rags 1 (GATOR1) complex, which is an important inhibitor of the mechanistic target of the rapamycin (mTORC1) pathway[22,23]. CRL3[KLHL22] recognizes and promotes polyubiquitination and degradation of DEPDC5 under nutrition-rich conditions[21].

Because the BTB domain is an oligomerization domain, the concept is well established that CRL3s exist and function as an oligomeric state, typically dimers[10–13]. The crystal structures of CUL3[NTD]-SPOP, CUL3[NTD]-KLHL11[BTB-BACK], CUL3[NTD]-KLHL3[BTB-BACK], CUL3[NTD]-A55[BTB-BACK], and CUL3[NTD]-KEAP1[BTB] have revealed how CUL3 binds to BTB-domain containing proteins[24–29]. Studies have shown that CRL3[KEAP1] recognizes NRF2 through DLG and ETGE motifs, which provide further convincing evidence that CRL3s function in a dimeric assembly state[30].

The assembly process of dimeric CRL3s occurs in two steps (Fig. 1a). In the first step, two monomeric BTB domain-containing proteins assemble into a dimeric core through an evolutionarily conserved BTB domain. To ensure that the first assembly step generates an active homodimeric BTB core rather than an inactive heterodimeric BTB core, this process is regulated by the SCF[FBXL17]-dependent dimerization quality control (DQC) mechanism[31,32]. In the second step, the dimeric BTB core recruits two CUL3-RBX1 subunits. Several groups of interactions mediate binding between BTB-domain containing proteins and CUL3, including the BTB domain interacts with α-helix2, α-helix4 and α-helix5 of CUL3, the 3-box interacts with α-helix5 of CUL3, and the BTB domain interacts with an extension of the N-terminus of CUL3 (aa 17–24)[24–29]. However, the assembly mechanism of dimeric CRL3s still need to be further investigated.

In this work, we determine the cryogenic electron microscopic (cryo-EM) structure of CRL3[KLHL22] at an overall resolution of 3.8 Å. We find that a conserved CUL3 N-terminal motif contributes to the dimeric assembly of CRL3[KLHL22]. We name this motif as the NA (N-terminal Assembly) motif for its important role in dimeric CRL3s assembly. Deletion of the CUL3 NA motif leads to decreased binding affinity and impairs assembly between CUL3 and KLHL22, which significantly impairs the E3 activity of CRL3[KLHL22]. In addition, we find that the CUL3 NA motif is evolutionarily conserved and participates in the assembly

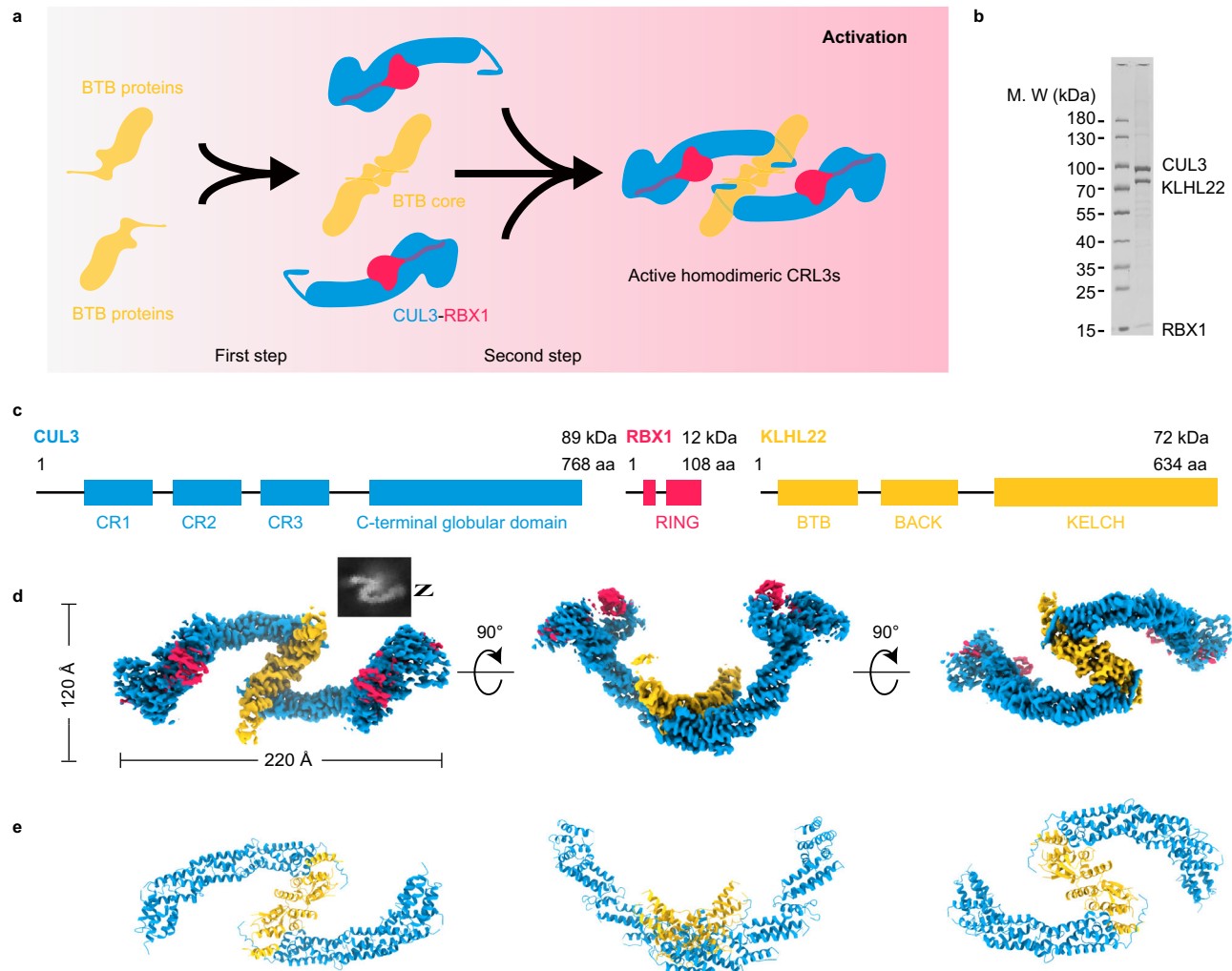

**Fig. 1 | Cryo-EM structure of dimeric human CRL3[KLHL22]. a** Shown are the two steps of dimeric CRL3s assembly. The first step is the formation of a dimeric BTB core. The second step is the recruitment of two CUL3-RBX1 subunits by the BTB core and the formation of dimeric CRL3s. **b** SDS-PAGE gel of purified CRL3[KLHL22]. The result shown is representative of at least three biological replicates. Source data are provided as a Source Data file. **c** Domain schemes of CUL3, RBX1, and KLHL22.

**d**, **e** Cryo-EM density map and cartoon model of dimeric CRL3[KLHL22]. The map was present with *C2* symmetry with dimensions indicated in the top view (left); the side view (middle), and the bottom view (right), are also shown (**d**). The corresponding models of CUL3 (blue), RBX1 (red), and KLHL22 (yellow) in cartoon representation are shown in the bottom panel (**e**).

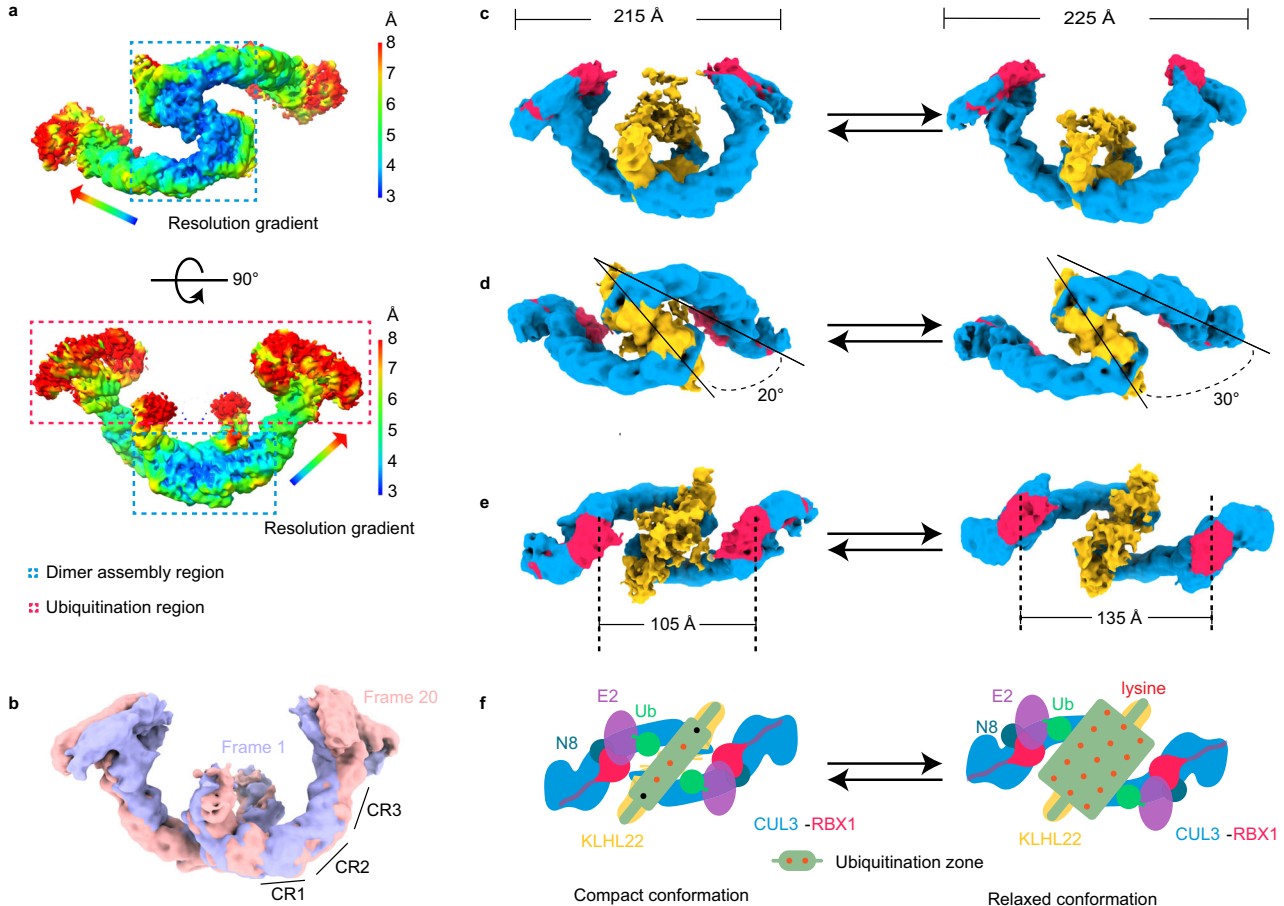

**Fig. 2 | Conformational dynamics of the dimeric CRL3^KLHL22 generate a variable ubiquitination zone. a** Local resolution map of CRL3^KLHL22. The resolution decreased from the dimer assembly region (blue dashed line) to the ubiquitination region (red dashed line). **b** Overlay of cryo-EM density maps with compact (Frame 1, purple) and relaxed (Frame 20, pink) conformations of CRL3^KLHL22. The dynamics of the CUL3 three helix-bundle repeats domain (CR1, CR2, and CR3) contribute to the conformational change. **c−e** CRL3^KLHL22 undergoes conformational changes from a compact conformation to a relaxed conformation. The long-axis distance increased from 215 Å to 225 Å (**c**). The angle between KLHL22 and CUL3 scaffold subunit increased from 20° to 30° (**d**). The distance between two RBX1 subunits increased from 105 Å to 135 Å (**e**). **f** Shown is a cartoon model of CRL3^KLHL22 conformational changes. The dynamic of the CUL3 scaffold subunit leads CRL3^KLHL22 to transition between a compact conformation and a relaxed conformation, which generates a variable ubiquitination zone.

of CRL3^KLHL12 and CRL3^KEAP1. Furthermore, our structure of dimeric CRL3^KLHL22 reveals that the dynamics of dimeric assembly leads CRL3^KLHL22 to a transition between a compact conformation and a relaxed conformation, which gives CRL3^KLHL22 a variable ubiquitination zone. Above all, we determined the cryo-EM structure of CRL3^KLHL22 and revealed that CUL3 N-terminal motif contributes to assembly of CRL3s. Our study provides insight into how the dimeric CRL3s are assembled, which may allow us to understand better the role of CRL3s in physiological and pathological processes.

## Results

### Cryo-EM structure of CRL3^KLHL22

To understand how dimeric CRL3s assemble in solution, we first obtained architectural evidence of the E3 ubiquitin ligase CRL3^KLHL22 using cryo-EM. We expressed in SF9 cells and then purified the full-length CUL3-RBX1-KLHL22 complex (Fig. 1b, Supplementary Fig. 1a, b). After optimization of the CUL3-RBX1-KLHL22 complex by Grafix[33] and examination by negative staining EM, a homogenous CUL3-RBX1-KLHL22 complex sample was obtained (Supplementary Fig. 1c−f). We then determined the cryo-EM structure of CRL3^KLHL22 at an overall resolution of 3.8 Å (Supplementary Fig. 2, Supplementary Table 1). The overall shape of CRL3^KLHL22 dimer resembles the letter 'Z' when

projected from the top (Fig. 1c, d). The two CUL3-RBX1 molecules consist of the upper and lower horizontal lines of the letter 'Z', and the dimeric KLHL22 project into the diagonal line. The two identical CUL3-KLHL22-RBX1 protomers consist of half of the 'Z'. Two CUL3-RBX1-KLHL22 protomers present in a symmetric unit forming a homo-dimeric assembly with an overall dimension of $220 \times 120 \times 120$ Å (Fig. 1d). Based on the high quality of the cryo-EM map in the N-terminal domain of CUL3, the BTB domain of KLHL22, together with the accurate prediction models of CUL3 and KLHL22 by Alphafold2[34,35], we built an atomic model of the CUL3-KLHL22 dimer (Fig. 1e).

### Conformational dynamics of dimeric assembly generate a variable ubiquitination zone

Based on functional differences, CRL3^KLHL22 may be divided into two functional regions (Fig. 2a). The first is the N-terminal dimer assembly region, including the N-terminal BTB domain of KLHL22 and the N-terminal Cullin Repeats (CR) domain of CUL3. The second functional region is the ubiquitination region, which includes the Kelch domain of KLHL22 and the C-terminal domain of CUL3 and RBX1. We found that the local resolution, from the N-terminal dimer assembly region to the ubiquitination region, was decreased (Fig. 2a), which suggested that the dimer assembly region is more rigid than the ubiquitination region.

As an E3 ubiquitin ligase, CRL3[KLHL22] conjugates the ubiquitin chain (both priming and elongation) to substrates with different sizes, such as PLK1, DEPDC5, and PD1[19–21]. To complete the ubiquitination process, the CRL3[KLHL22] must undergo conformational changes. To further investigate this dynamic process, we performed the 3D variability analysis. This showed that CRL3[KLHL22] transitioned from a compact conformation to a relaxed conformation (Fig. 2b–f, Supplementary Movie 1). The length of the long-axis of CRL3[KLHL22] increased from 215 Å to 225 Å (Fig. 2c), and the angle between the substrate receptor KLHL22 and the CUL3 scaffold subunit increased from 20° to 30° (Fig. 2d). The conformational changes were induced mainly by the three CR domains of CUL3 (Fig. 2b). These curved CR domain repeats provide the structural basis for the rotation of the CUL3 scaffold, although it has been commonly considered as a rigid scaffold[36]. The distance between the two RBX1 subunits also increased from 105 Å to 135 Å (Fig. 2e). Compared to the compact conformation, the relaxed conformation has a more spacious ubiquitination zone (Fig. 2f). We note that similar CRL3[KLHL22] conformational changes were also observed in a very recent study by using a cryo-EM dataset of substrate-bound CRL3[KLHL22]-GDH1 complex[37]. Taken together, the dynamics of the dimeric assembly of CRL3[KLHL22] generate a variable ubiquitination zone, which may provide the structural basis for the ubiquitination of substrates with different molecular sizes and attach ubiquitin to different lysine residues within the substrates.

## KLHL22 dimer core assembly mainly mediated by hydrophobic interaction

The first step of dimeric CRL3[KLHL22] assembly is KLHL22 dimerization through its BTB domain which is similar to the classical BTB domain[14,24–29] (Fig. 3a). In addition to the core BTB fold (α-helix A1–A5 and β-strands B1–B3), the BTB domain of KLHL22 also has an α1 helix and domain-swapping β1 strands (Fig. 3a). The KLHL22 dimer formation is mediated mainly through hydrophobic interactions (Fig. 3b, c) which are conserved among the BTB domain-containing protein family members, such as the SPOP, KLHL11, KLHL3, KEAP1, A55 dimer[24–29]. Following the core BTB fold is the 3-box, a pair of helices (A6 and A7), which is crucial for its interaction with CUL3[24–29].

The core BTB fold and α-helix A6 of the 3-box of KLHL22 aligned well with the crystal structure of KLHL11 (Supplementary Fig. 3a). However, we found that the α-helix A7 of the 3-box had a 30° rotation compared with KLHL11 (Supplementary Fig. 3a), which suggested that the BACK domain of KLHL22 may exhibit a different orientation than that of KLHL11. Due to the dynamic of the BACK domain, we were not able to determine its complete structure. However, based on the first three α-helices of the BACK domain, we found that the BACK domain of KLHL22 was rotated 52° compared with the KLHL11 BTB-BACK domain (Supplementary Fig. 3b). Similar conformational dynamics of CRL3[KLHL22] were also observed by analysis of the cryo-EM dataset prepared for substrate-bound CRL3[KLHL22]-GDH1 complex[37]. The overall shape of the KLHL22 BTB-BACK dimer resembled a 'fluttering bird' conformation with each BTB-BACK monomer comprising one side of the 'wing'. This differed from the overall shape of the KLHL11 BTB-BACK dimer and NPR1, which was more like a 'gliding bird' (Supplementary Fig. 3b)[24,38].

## CUL3 NA motif participates in the dimeric CRL3[KLHL22] assembly

The second step of dimeric CRL3[KLHL22] assembly is the recruitment of two CUL3-RBX1 subunits by the KLHL22 dimer. Previous studies showed that the recruitment of CUL3-RBX1 by BTB domain-containing proteins can occur through three groups of interactions: (1) the BTB domain interacts with α-helix2, α-helix4 and α-helix5 of CUL3, (2) the 3-box interacts with α-helix5 of CUL3, and (3) the BTB domain interacts with an extension of the N-terminus of CUL3 (aa 17–24)[24–29]. To further investigate how dimeric CRL3[KLHL22] is assembled, we generated a mask for local refinement of the N-terminal dimer assembly region, and the

resolution was optimized to 3.3 Å (Supplementary Fig. 2a). We observed an extra density surrounds KLHL22 (Fig. 3d). Considering that CUL3 aa 17–24 has been observed in the crystal structure of CUL3[NTD]-KLHL11[BTB-BACK] [24] and based on the density map, we speculated that this extra density also belonged to the CUL3 N-terminal residues (Fig. 3d). Here, we refer to the CUL3 N-terminal sequence (1–24) as the NA motif (N-terminal Assembly motif) because of its important role in CRL3s assembly in the following description.

We found that the CUL3 NA motif, together with the CUL3 CR1 domain, tightly wraps the KLHL22 (Fig. 3d). The CR1 domain interacts with the BTB domain of KLHL22 over a wide area mainly through polar interactions (Fig. 3e). A well-defined hydrophobic core formed between Leu52, Phe54, Leu57, Ile122 of CUL3 and Leu83 of KLHL22 also plays a role in CUL3-RBX1 recruitment (Fig. 3f). Previous studies have identified a conserved Φ-x-E motif within CUL3 adapters, where Φ represents a hydrophobic residue, are crucial for SPOP binding to CUL3[29]. The KLHL22-Leu83 exactly follows the conserved Φ-x-E motif pattern and corresponds to the hydrophobic Φ residue (Supplementary Fig. 3c), which was also observed in the crystal structures of CUL3[NTD]-SPOP, CUL3[NTD]-KLHL11[BTB-BACK], CUL3[NTD]-KLHL3[BTB-BACK], CUL3[NTD]-A55[BTB-BACK], and CUL3[NTD]-KEAP1[BTB] (Supplementary Fig. 3d)[24–28].

The binding of the CUL3 NA motif to KLHL22 is mediated by multiple types of interactions, including hydrophobic interaction and numerous electrostatic attractions (Fig. 3g, h). The hydrophobic residues of the CUL3 NA motif, such as Met23, Phe21, Ile18, and Met16, project onto a hydrophobic surface of KLHL22 formed by Leu161, Phe162, Leu158, and Val155. Hydrogen bonds are involving the side chains of the CUL3 NA motif residues (Arg19, Arg17) and the KLHL22 residues from protomer 1 (Gln121, Glu122, Asp148, W146, Arg157). Although the density corresponding to aa 14–24 is clear enough (Fig. 3g), the density corresponding to aa 1–13 of CUL3 is blurry, which suggests greater dynamics of CUL3 aa 1–13. By lowering the contour level, we observed additional density parallel to the domain-swapping β1' strand of KLHL22 promoter 2, which likely corresponds to the CUL3 aa 1–13 (Fig. 3i, j). A similar map density was also observed in a recent study (Fig. 3k)[37]. To further investigation of how the aa 1–13 of CUL3 interacts with KLHL22, we performed molecular dynamic simulation analysis. Within a 1 μs simulation, we found that aa 1–13 interacted with the domain-swapping β1' strand of KLHL22 protomer 2, which is consistent with the observed extra map density (Fig. 3l, and Supplementary Fig. 3e, f). Together, these results suggest that both the CR1 domain and the complete CUL3 N-terminal sequence participate in the assembly of dimeric CRL3[KLHL22].

## CUL3 NA motif deletion impairs dimeric assembly of CRL3[KLHL22]

To investigate the role of the CUL3 NA motif in the assembly of CRL3[KLHL22], we generated two truncated mutants: CUL3[Δ1–24], in which the NA motif was deleted entirely, and a partial deletion mutant CUL3[Δ1–13], which was expected to impair the interaction between CUL3 and the domain swapping β-sheet of KLHL22 (Fig. 4a). We first investigated the contribution of the CUL3 NA motif to binding affinity using isothermal titration calorimetry (ITC) assay. As the full-length KLHL22 alone was not properly folded in our recombinant protein expression system, KLHL22[1–178] was used in the assay (Supplementary Fig. 4a, b). The results showed that CUL3 binds to KLHL22[1–178] with a binding affinity of approximately 27 nM, while CUL3[Δ1–13] and CUL3[Δ1–24] bind to KLHL22[1–178] with binding affinities of approximately 87 nM and 383 nM, respectively (Fig. 4b–d, and Supplementary Fig. 4c–e). These 3.2-fold and 14.2-fold decreases in binding affinity suggest that the complete CUL3 NA motif is important for recruiting CUL3-RBX1 to the KLHL22 dimer. To investigate whether the CUL3 NA motif is also critical for the association of CUL3 and KLHL22 in cells, we performed in vivo pull-down assays. The results showed that deletion of the CUL3 NA motif (as in CUL3[Δ1–24]) significantly decreases the interaction between CUL3-

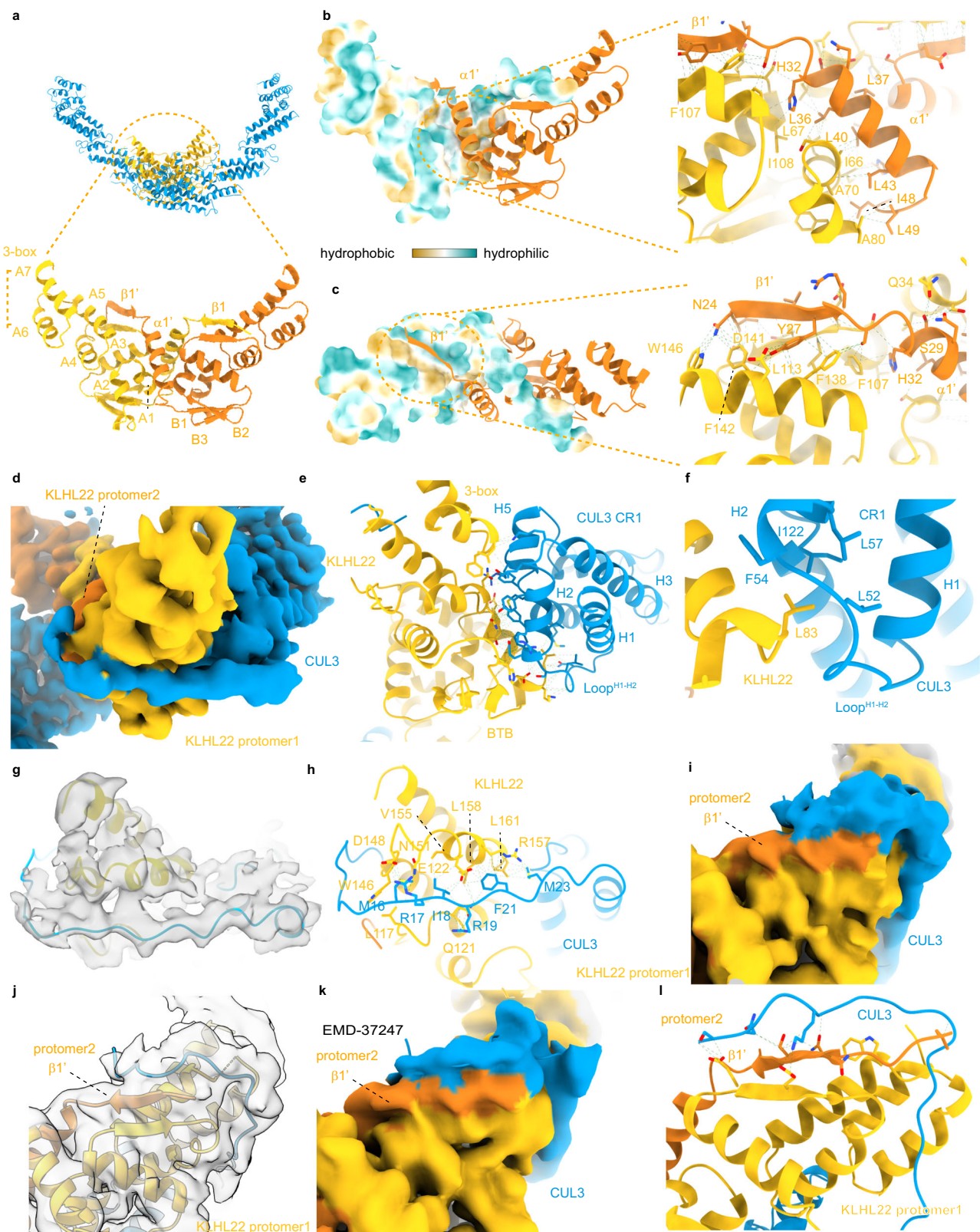

RBX1 and KLHL22 (Fig. 4e, lane 1 vs lane 3, and Supplementary Fig. 4f). Even a partial CUL3 NA motif deletion (as in CUL3$^{\Delta 1-13}$) significantly reduces the interaction between CUL3-RBX1 and KLHL22 (Fig. 4e, lane 1 vs lane 2, and Supplementary Fig. 4f). These observations are consistent with the results of the ITC assay. In addition, we found that the N-terminal extension of KLHL22 does not significantly impact its interaction with CUL3 (Fig. 4e, lane 1 vs lane 4). These results suggest

that the CUL3 NA motif is important for recruiting CUL3-RBX1 subunits to KLHL22.

To investigate whether the CUL3 NA motif plays a role in the assembly of dimeric CRL3$^{KLHL22}$, we analyzed the molecular weight of the CUL3-RBX1-KLHL22$^{1-178}$ using size-exclusion chromatography (SEC). KLHL22$^{1-178}$ was preincubated with CUL3$^{WT}$, CUL3$^{\Delta 1-13,}$ or CUL3$^{\Delta 1-24}$ at molar ratios of 1:1, followed by SEC analysis. The elution

**Fig. 3 | Both the BTB domain and the CUL3 N-terminal motif participate in dimeric CRL3^KLHL22 assembly. a** CRL3^KLHL22 dimer with the BTB domain enlarged. The cartoon model shows the core BTB fold (A1-A5, B1-B3), the 3-box (A6 and A7), and the inter-model swapped α1 and β1. BTB domain from protomer 1 and protomer 2 are colored yellow and orange, respectively. **b, c** Hydrophobic interactions mediate BTB dimer formation. Hydrophobic residues located within α1′ (**b**) and β1′ (**c**) from KLHL22 protomer 2 contact with hydrophobic residues from protomer 1 (shown in surface representation in the left panel). **d** Cryo-EM density map of the CUL3 NA motif (blue) interacts with KLHL22 (yellow). **e** Shown is a large area of interaction between the BTB domain of KLHL22 and the H2, H5, and Loop^H1-H2 of CUL3. The interaction is mainly mediated through polar interactions.

**f** Hydrophobic core formed by the BTB domain of KLHL22 and the CR1 domain of CUL3. Inter-facial residues (Leu83 from KLHL22; Leu52, Phe54, Leu57, Ile122 from CUL3) are shown. **g** Overlay of CUL3 NA motif cartoon model (aa 14–24) on the density map. **h** The molecular interaction between CUL3 NA motif (aa 14–24) and KLHL22. **i, j** Cryo-EM density map of the CUL3 NA motif (blue) interacts with KLHL22 promoter 2 (orange). Overlay of CUL3 NA motif cartoon model on the density map (**j**). **k** Cryo-EM density map (EMD-37247) of the CUL3 NA motif (blue) interacts with KLHL22 promoter 2 (orange). **l** The molecular interaction between the CUL3 NA motif (aa 2–13) and the domain swapping β sheet of KLHL22. The interactions were revealed by 1 μs molecular dynamics simulation.

volume of the CUL3^WT-RBX1-KLHL22^1–178 is 10.1 ml (Fig. 4f, Supplementary Fig. 5a). However, the elution volumes of the CUL3^Δ1–13-RBX1-KLHL22^1–178 and CUL3^Δ1–24-RBX1-KLHL22^1–178 are 10.7 and 10.8 ml, respectively (Fig. 4f, Supplementary Fig. 5b). These results suggested that the molecular weights of the deletion mutant CUL3^Δ1–13-RBX1-KLHL22^1–178 and CUL3^Δ1–24-RBX1-KLHL22^1–178 are smaller than the wild-type CUL3^WT-RBX1-KLHL22^1–178.

We also measured the molecular weight of CUL3^WT-RBX1-KLHL22^1–178 and CUL3^Δ1–24-RBX1-KLHL22^1–178 using multi-angle light scattering coupled with SEC (SEC-MALS). The SEC-MALS analysis of CUL3^WT-RBX1-KLHL22^1–178 showed a molecular weight of 241 kDa, which is consistent with a dimeric complex (calculated molecular weight 245 kDa, of one dimeric KLHL22^1–178 binding with two CUL3^WT-RBX1) (Fig. 4g). However, The SEC-MALS analysis of the CUL3^Δ1–24-RBX1-KLHL22^1–178 showed two protein peaks. The first indicated a molecular weight of 155 kDa, which corresponds to one dimeric KLHL22^1–178 (calculated molecular weight 41 kDa, Supplementary Fig. 5c) binding with one CUL3^Δ1–24-RBX1 (calculated molecular weight 100 kDa) (Fig. 4h). The second protein peak showed a molecular weight of 102 kDa, which corresponds to excessive unbound CUL3^Δ1–24-RBX1 (Fig. 4h). Similar results were also obtained by using the mass photometry assay (Supplementary Fig. 5d–i). Thus, these results again indicate that the CUL3 NA motif participates in the assembly of dimeric CRL3^KLHL22.

To provide further evidence that deletion of the CUL3 NA motif impairs assembly of CRL3^KLHL22, we resolved the cryo-EM structure of the CUL3^Δ1–24-RBX1-KLHL22^1–178 with 3.9 Å resolution (Supplementary Fig. 6a, Supplementary Table 1). Rigid body-fitting of the CUL3^Δ1–24-RBX1 and KLHL22^1–178 structure confirms that the complex contains one dimeric KLHL22^1–178 and one CUL3^Δ1–24-RBX1 (Fig. 4i). We also found that about 6% (5070/79349, Supplementary Fig. 6) of particles exist as dimer states, suggesting that the KLHL22 dimer can also recruit two CUL3^Δ1–24-RBX1 molecules. We speculate that the CUL3^Δ1–24-RBX1-KLHL22^1–178 may exist in a state of dynamic equilibrium between the (CUL3^Δ1–24-RBX1)₂-(KLHL22^1–178)₂ state and the (CUL3^Δ1–24-RBX1)₁-(KLHL22^1–178)₂ state, with the equilibrium favoring the formation of (CUL3^Δ1–24-RBX1)₁-(KLHL22^1–178)₂ due to decreased binding affinity between KLHL22 and CUL3^Δ1–24-RBX1 (Supplementary Fig. 6b). To investigate whether adding more CUL3^Δ1–24-RBX1 would drive this equilibrium toward the direction of forming more (CUL3^Δ1–24-RBX1)₂-(KLHL22^1–178)₂ complex, we incubated KLHL22^1–178 with CUL3^Δ1–24-RBX1 at molar ratio of 1:2 or 1:4, respectively. The results of analytical SEC show that adding more CUL3^Δ1–24-RBX1 does not push this equilibrium toward the direction of forming more (CUL3^Δ1–24-RBX1)₂-(KLHL22^1–178)₂ complex (Fig. 4j and Supplementary Fig. 5j–l). Given that the presence of (CUL3^Δ1–24-RBX1)₂-(KLHL22^1–178)₂ complex in our cryo-EM data, we speculate that the reason why adding more CUL3^Δ1–24-RBX1 does not push this equilibrium toward the direction of forming more (CUL3^Δ1–24-RBX1)₂-(KLHL22^1–178)₂ complex may be attributed to a fast off-rate between CUL3^Δ1–24-RBX1 and KLHL22^1–178, and the (CUL3^Δ1–24-RBX1)₂-(KLHL22^1–178)₂ complex undergoes dissociation within the column. Taken together, these results again indicate that the CUL3 NA motif is important for the assembly of the dimeric CRL3^KLHL22.

## Evolutionarily conserved CUL3 NA motif contributes to the assembly and E3 activity of other CRL3s

As mentioned earlier, there are approximately 183 BTB domain-containing proteins in human cells and the core BTB fold is evolutionarily conserved[14]. We determined that the residues of the CUL3 NA motif are also evolutionarily conserved (Fig. 5a). Thus, we wished to know if the CUL3 NA motif also contributes to the assembly of other CRL3s. For this reason, we undertook a further analysis of CRL3^KLHL12, an important regulator of COPII coat formation[39]. Consistent with the results obtained for CRL3^KLHL22, the elution volumes of CUL3^Δ1–24-RBX1-KLHL12^1–161 (11.5 ml) and CUL3^Δ1–13-RBX1-KLHL12^1–161 (11.5 ml) were greater than CUL3^WT-RBX1-KLHL12^1–161 (10.4 ml), which suggested that the molecular sizes of CUL3^Δ1–24-RBX1-KLHL12^1–161 and CUL3^Δ1–13-RBX1-KLHL12^1–161 were smaller than that of CUL3^WT-RBX1-KLHL12^1–161 (Fig. 5b, Supplementary Fig. 7a, b). Similar results were observed for CRL3^KEAP1 (Fig. 5c, Supplementary Fig. 7c–e). SEC-MALS analysis showed that the molecular weight of CUL3^WT-RBX1-KLHL12^1–161 is 239 kDa, which corresponds to the dimeric state (calculated molecular weight 240 kDa, Fig. 5d). The molecular weight of CUL3^Δ1–24-RBX1-KLHL12^1–161 is 135 kDa (Fig. 5e), which corresponds to one dimeric KLHL12^1–161 (calculated molecular weight 36 kDa, Supplementary Fig. 7f) binding with one CUL3^Δ1–24-RBX1 (calculated molecular weight 100 kDa). In addition, consistent with the results of KLHL22, adding more CUL3^Δ1–24-RBX1 subunits does not drive the formation of more (CUL3^Δ1–24-RBX1)₂-(KLHL12^1–161)₂ and (CUL3^Δ1–24-RBX1)₂-KEAP1₂ complex (Fig. 5f, g and Supplementary Fig 7g–l). Together, these results indicate that the CUL3 NA motif-contributed assembly may be conserved among CRL3s.

To further investigate the important roles of the CUL3 NA motif for the E3 activity of CRL3s, we performed the ubiquitination assays. The results showed that deletion of the CUL3 NA motif decreased the polyubiquitination level of DEPDC5 (Supplementary Fig. 8a, lane 4 vs lane 6). Partial deletion of the CUL3 NA motif (aa1–13) also impairs the E3 enzyme activity of CRL3^KLHL22 (Supplementary Fig. 8a, lane 4 vs lane 5). Furthermore, ubiquitination of NRF2 by CRL3^KEAP1 observed by an in vitro ubiquitination assay showed similar results (Fig. 5h, Supplementary Fig. 8b). Together, our results suggest that the CUL3 NA motif is important for the assembly and E3 ligase activity of CRL3s (Fig. 5i).

## Discussion

The concept that CRL3s function as oligomer states, typically dimers, has been well established in the past two decades. To further investigate the assembly mechanism of CRL3s, we determined the cryo-EM structure of CRL3^KLHL22, revealing that the conserved CUL3 NA motif is important for both the dimeric assembly and the E3 ligase activity of CRL3^KLHL22. We demonstrated that complete or partial deletion of the CUL3 NA motif result in impaired dimeric CRL3^KLHL22 assembly and E3 ligase activity. Furthermore, we found that CRL3^KLHL12 and CRL3^KEAP1 employ the same strategy to ensure the integrity of dimeric assembly, indicating that this assembly strategy may be conserved among CRL3s. Additionally, our research uncovered that the dynamics of the dimeric assembly of CRL3^KLHL22 create a variable ubiquitination zone,

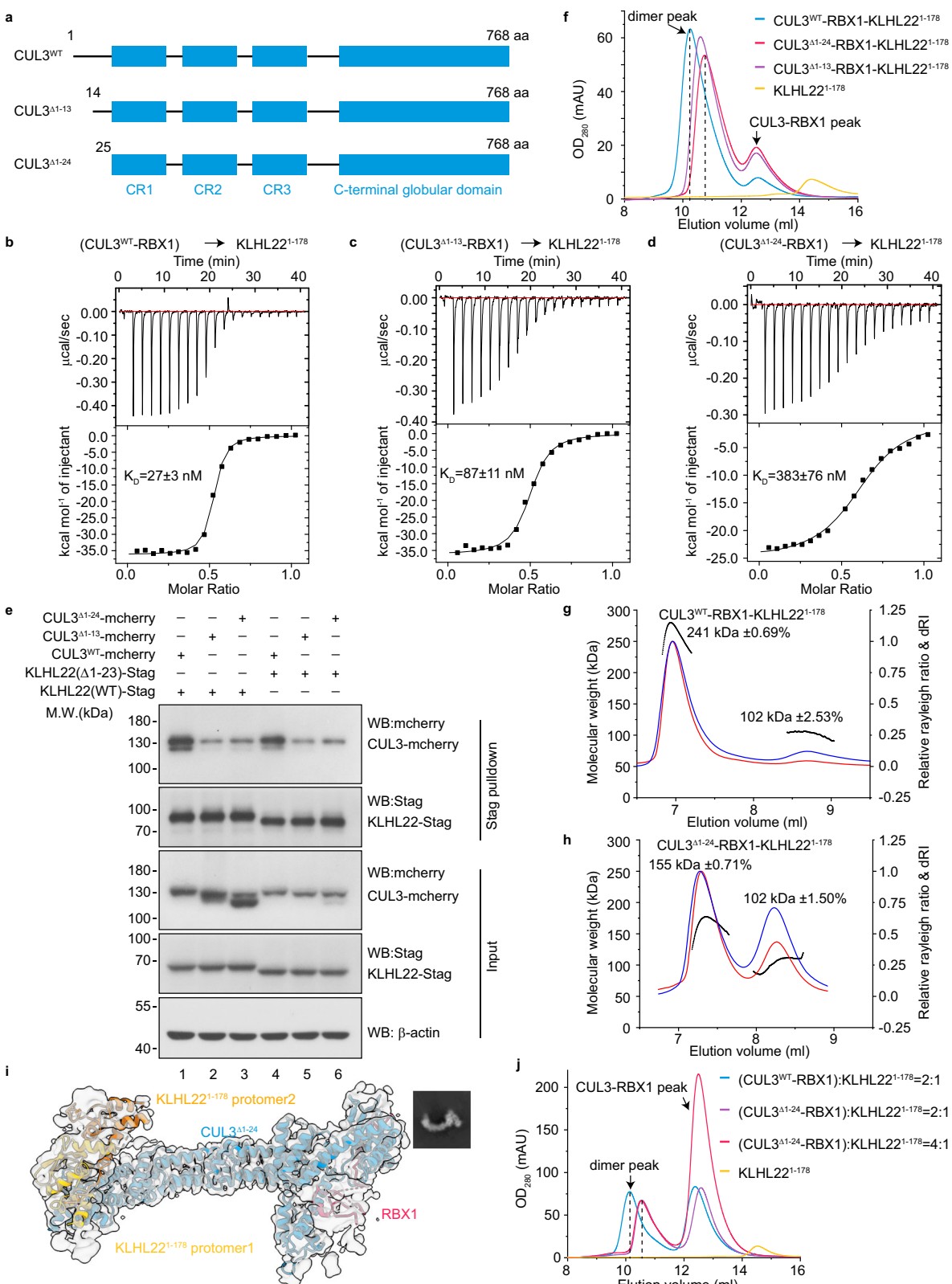

potentially facilitating recognition and ubiquitination of substrates. Overall, our study provides important molecular and mechanistic insights into the assembly of dimeric CRL3s.

CRLs are multi-subunit E3 complexes, with each subunit contributing to different functions. For instance, the RBX1/2 subunit is important for E2 docking, the substrate receptor subunits are important for substrate recognition and the Cullin proteins function as scaffold subunits. The Cullin scaffold subunit contains three five-helix bundle repeats[36], which function like a spring. The 3D variability analysis of CRL3[KLHL22] shows that the elasticity of the CUL3 scaffold subunit leads to changes in the long axis of CRL3[KLHL22] from 215 Å to 225 Å, and changes in the angle between the substrate receptor KLHL22 and the C-terminal globular domain of CUL3 and RBX1 (from 20° to 30°). This dynamic behavior generates a variable ubiquitination zone in CRLs

**Fig. 4 | CUL3 NA motif deletion impairs dimeric assembly of CRL3$^{KLHL22}$.**
**a** Diagram depicting domain organization of CUL3 and truncation mutants used in subsequent S-tag pulldown, SEC, ITC, and SEC-MLAS assays. CR, Cullin repeats. **b–d** ITC measurement of binding affinity between KLHL22$^{1–178}$ and CUL3$^{WT}$ (**b**), CUL3$^{Δ1–13}$ (**c**), and CUL3$^{Δ1–24}$ (**d**). The binding constants (KD ± SD) are indicated. This experiment performed once. **e** Deletion of CUL3 NA motif impairs interaction between CUL3 and KLHL22 in intact cells. KLHL22$^{WT}$-(S-tag) or KLHL22$^{Δ1–23}$-(S-tag) were co-expressed with CUL3$^{WT}$-mCherry, CUL3$^{Δ1–13}$-mCherry, and CUL3$^{Δ1–24}$-mCherry, respectively, in HEK 293T cells, and S-tag pulldown assays were performed. This S-tag pulldown assay performed once. **f** SEC analysis of the effect of CUL3 NA motif deletion on the assembly of the CUL3-RBX1-KLHL22$^{1–178}$ complex. Chromatograms show the elution profiles of CUL3$^{WT}$-RBX1-KLHL22$^{1–178}$ (blue line), CUL3$^{Δ1–13}$-RBX1-KLHL22$^{1–178}$ (purple line), CUL3$^{Δ1–24}$-RBX1-KLHL22$^{1–178}$ (red line), and KLHL22$^{1–178}$ alone (yellow line). Peaks corresponding to dimers and excess unbound CUL3-RBX1 are indicated with dark arrows. The results shown are representative of three biological replicates. **g, h** SEC-MALS analysis of the effect of CUL3 NA motif deletion on the assembly of CUL3-RBX1-KLHL22$^{1–178}$ complex. **g** and **h** Show the elution profiles of CUL3$^{WT}$-RBX1-KLHL22$^{1–178}$ (**g**) and CUL3$^{Δ1–24}$-RBX1-KLHL22$^{1–178}$ (**h**), respectively. This experiment performed two times. **i** Cryo-EM density map of the CUL3$^{Δ1–24}$-RBX1-KLHL22$^{1–178}$ complex. The KLHL22$^{1–178}$ dimer binds only one CUL3$^{Δ1–24}$-RBX1. The corresponding 2D classification of the CUL3$^{Δ1–24}$-RBX1-KLHL22$^{1–178}$ complex is shown in the upper right corner. **j** SEC analysis of the effect of the concentration of CUL3-RBX1 on the assembly of the CUL3$^{Δ1–24}$-RBX1-KLHL22$^{1–178}$ complex. Chromatograms show the elution profiles of CUL3$^{WT}$-RBX1-KLHL22$^{1–178}$ (blue line, molar ratios 2:1), CUL3$^{Δ1–24}$-RBX1-KLHL22$^{1–178}$ (purple line, molar ratios 2:1), CUL3$^{Δ1–24}$-RBX1-KLHL22$^{1–178}$ (red line, molar ratios 4:1), and KLHL22$^{1–178}$ alone (yellow line). This experiment performed once. For **b–h** and **j**, source data are provided as a Source Data file.

which is needed to ubiquitylate different substrates with various lengths of ubiquitin chains.

E3 ubiquitin ligases are known to adopt different strategies to expand their ubiquitination zones. The conformational flexibility of the HECT domain and substrate binding domain facilitates the ubiquitination process of the HECT type E3 ubiquitin ligase[40]. In the case of the dimeric Fbxw subfamily of CRL1s, the intraprotomer gap and interprotomer gap within the CRL1$^{Cdc4}$ create two distinct substrate-to-catalytic site separation distances. This variation allows SCF$^{Cdc4}$ to establish a variable ubiquitination zone, facilitating efficient ubiquitin priming and elongation at different lysine receptor site within substrate[41]. In this study, we found that the dynamics of the assembly of CRL3$^{KLHL22}$ lead to transitions between the compact and the relaxed conformations. This dynamic property primarily results from the elasticity of three-helix bundle repeats of the CUL3 scaffold subunit. As other Cullin members (CUL1, CUL2, CUL3, CUL4A, CUL4B, CUL5, and CUL7) contain three five-helix bundle repeats, we hypothesize that the Cullin subunit may not only function as a scaffold subunit but also help to provide variable ubiquitination zones for the CRLs.

It has long been a puzzle as to how CRL3s assemble and maintain their functional dimeric assembly state. Dimerization is a common process for the functional activation of proteins. For example, the dimerization of CRL1$^{Fbx4}$ is required for substrate binding and E3 ligase activation[42]. Furthermore, the giant E3 ubiquitin ligase BIRC6 [baculoviral IAP repeat (BIR) domain-containing proteins 6] is assembled and functions as dimer[43–46]. The dimeric architecture of BIRC6 is important for binding of its substrates, including SMAC (Second mitochondria-derived activator of caspases) and serine protease HTRA2 (High temperature requirement protein A2). Moreover, the tumor suppressor PTEN exists as a dimer, which is important for its tumor suppressive activity[47,48]. Heterodimerization of wild-type and mutant PTEN proteins leads to a loss of function and causes a dominant-negative effect[47]. Similar dominant-negative effects also exist in the BTB domain-containing protein families[14]. Given that the BTB fold core is evolutionarily conserved, it is possible that BTB domain-containing proteins exist as heterodimers in cells[31,49]. Previous studies have shown that the SCF$^{FBXL17}$-dependent dimerization quality control mechanism is important for distinguishing heterodimer BTB domain-containing proteins from active homodimer BTB domain-containing proteins[31,32].

In this study, we found that the CUL3 NA motif interacts with both KLHL22 subunits, which suggested that CUL3 itself may participate in the assembly of dimeric CRL3$^{KLHL22}$. Further investigation showed that deletion of the CUL3 NA motif led to impaired dimeric assembly of CRL3$^{KLHL22}$ due to decreased binding affinity between KLHL22 and CUL3. The amino acid residues of the CUL3 NA motif are evolutionarily conserved, consistent with our observations that the CUL3 NA motif was also important for the dimerization of CRL3$^{KLHL12}$ and CRL3$^{KEAP1}$. These findings indicate that the CUL3 NA motif contributes to assembly is likely evolutionarily conserved within the CRL3s. More importantly, our ubiquitination assay demonstrated that the CUL3 NA motif deletion significantly impairs the E3 ligase activity of both CRL3$^{KLHL22}$ and CRL3$^{KEAP1}$, underscoring the important role of the CUL3 NA motif in the E3 ligase activity of dimeric CRL3s.

Once the functional dimeric assembly is formed, the question arises: how can this functional state maintained? Our 3D variability analysis showed that the spring-like three five-helix bundles of CUL3 can be stretched to generate a variable ubiquitination zone. This stretching, however, generates tensions that may affect the stability of dimeric CRL3s. Anchoring of KLHL22 by the CUL3 NA motif through multiple interactions may overcome these tensions to help maintain the stability and integrity of dimeric assembly of CRL3s.

In addition to the CRL3s that function as dimeric states, certain members of CRL1s can also assemble and function as dimeric states, notably within the Fbxw subfamily and Fbxo subfamily[41,42]. Biochemical assays have confirmed that dimerization is absolutely required for the ubiquitination of Pin2 by CRL1$^{Fbx4}$. The process of how CRL1s assemble into and maintain their dimeric state remains largely unknown. CUL1 also possesses a short N-terminal motif and an extra loop between α-helix 2 and α-helix 3. Whether this short CUL1 N-terminal motif and extra loop function similarly to the CUL3 NA motif, which plays a role in the dimeric CRL3s assembly, requires further investigation. Additionally, several other Cullin members feature N-terminal sequences. For instance, CUL4A and CUL4B have a long N-terminal sequence, which is important for recruitment of the adapter subunit DDB1[4,6]. Given that oligomeric assembly represents a common strategy for the functional regulation of CRLs, such as CRL4A$^{DCAF1}$ exists in both tetrameric and dimeric forms[50], we postulate that other CRLs may employ a similar strategy, akin to the CUL3 NA motif-contributed assembly, to regulate E3 ligase activity. Furthermore, neddylation can also regulate the assembled oligomeric state of CRLs. For instance, neddylation can switch the autoinhibited tetrameric CRL4$^{ADCAF1}$ into an activated dimeric state[50]. Considering that CRL3s exist not only in the dimeric state but also in higher oligomeric state, such as oligomeric SPOP, pentameric CRL3$^{KCTD5}$, tetrameric and octameric CRL3$^{KBTBD2}$ [51–53], it is worth to further investigating whether the CUL3 NA motif alone or together with neddylation can affect the higher oligomer state of CRL3s.

## Methods

### Cloning

All cDNAs (unless noted otherwise) used in this study were of *Homo sapiens* origin, and generated from mRNAs extracted from HEK 293T cells. For co-expression of CUL3 and RBX1, full-length CUL3 with a C-terminal His$_6$ tag and RBX1 without a tag were cloned into ORF1 and ORF2 of pFastbac-dual vector, respectively. Full-length KLHL22 and KEAP1 were cloned into pFastbac-htb vector with an N-terminal His$_6$ tag.

For biochemical studies, proteins were produced by cloning the coding sequence of KLHL22$^{1–178}$, KLHL12$^{1–161}$, and NRF2$^{1–100}$ into a

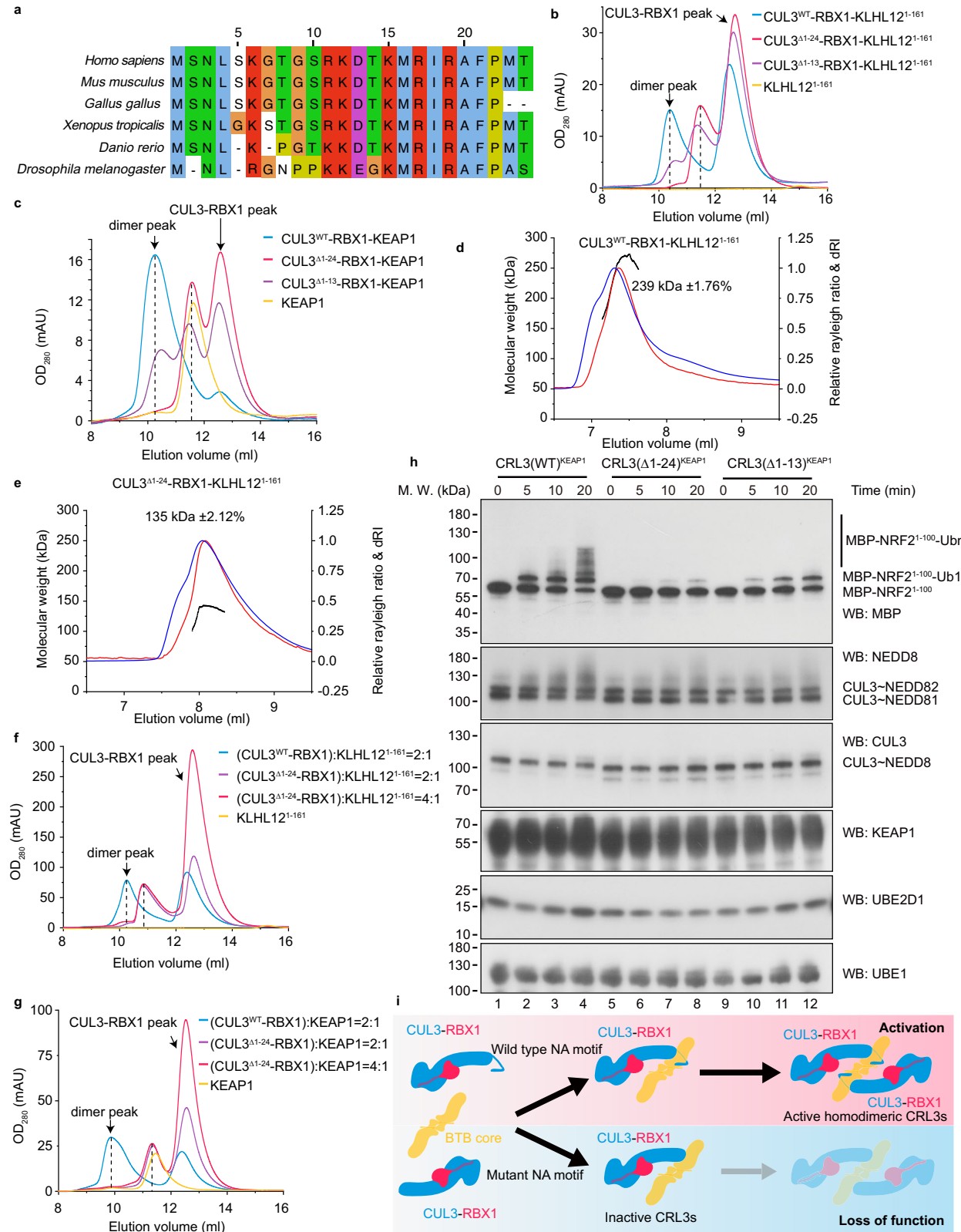

modified pET28a (+) vector with an N-terminal His$_6$-MBP tag and a TEV protease cleavage site (ENLYFQG).

For experiments in intact cells, full-length CUL3 with a C-terminal mCherry tag and RBX1 without tag were cloned into ORF1 and ORF2 of a pFastbac-dual vector, respectively, which had been modified to express in mammalian cells by inserting a CMV enhancer and promoter sequence before ORF1, and an EF1a sequence before ORF2. Full-length

KLHL22 with a C-terminal S-tag was cloned into the pCDH-puro vector. Full-length DEPDC5 with a C-terminal mCherry tag was cloned into ORF1 of the modified pFastbac-dual vector as described above.

CUL3$^{\Delta1-13}$ and CUL3$^{\Delta1-24}$ deletion mutants were generated by using PCR-based site-directed mutagenesis using wild type CUL3 cDNA as template. The integrity of all inserted sequences or truncations was verified by sequencing.

**Fig. 5 | CUL3 NA motif-contributed assembly is conserved in CRL3$^{KLHL12}$ and CRL3$^{KEAP1}$. a** Evolutionary conservation analysis showing the conserved amino acid residues within the CUL3 NA motif. Conserved residues are colored. **b** SEC analysis of the effect of CUL3 NA motif deletion on the assembly of the CUL3-RBX1-KLHL12$^{1–161}$ complex. The chromatograms show the elution profiles of CUL3$^{WT}$-RBX1-KLHL12$^{1–161}$ (blue line), CUL3$^{Δ1–13}$-RBX1-KLHL12$^{1–161}$ (purple line), CUL3$^{Δ1–24}$-RBX1-KLHL12$^{1–161}$ (red line), and KLHL12$^{1–161}$ alone (yellow line). This experiment performed at least three times. **c** SEC analysis of the effect of CUL3 NA motif deletion on the assembly of the CUL3-RBX1-KEAP1 complex. This experiment performed at least three times. **d, e** SEC-MALS analysis of the effect of CUL3 NA motif deletion on the assembly of the CUL3-RBX1-KLHL12$^{1–161}$ complex. **d** and **e** Show the results of CUL3$^{WT}$-RBX1-KLHL12$^{1–161}$ (**d**) and CUL3$^{Δ1–24}$-RBX1-KLHL12$^{1–161}$ (**e**), respectively. This experiment performed once. **f, g** SEC analysis of the effect of CUL3 NA motif deletion and the concentration of CUL3-RBX1 on the assembly of the CUL3-RBX1-KLHL12$^{1–161}$ (**f**) and CUL3-RBX1-KEAP1 (**g**). This experiment performed once. **h** E3 ligase activity analysis using an in vitro ubiquitination assay. Western blots show the impaired E3 activity of CRL3$^{KEAP1}$ toward MBP-NRF2$^{1–100}$ associated with the deletion of the CUL3 NA motif. This experiment performed once. **i** Shown is a cartoon model of CUL3 NA motif contributes to assembly of CRL3s. In the second step of the assembly process of CRL3s, the BTB core (or with substrate) recruits two CUL3-RBX1 subunits and forms active dimeric CRL3s (up panel). Deletion of the CUL3 NA motif or a disease-related mutation within the CUL3 NA motif decreases the binding affinity between the CUL3 and KLHL22, which in turn impairs the integrity and E3 ligase activity of the CRL3s (bottom panel). For **b–h**, source data are provided as a Source Data file.

Primers used for plasmids construction list in Supplementary Table 2.

## Cell culture and transfections

HEK 293T cells were purchased from Thermo Fisher Scientific and transient transfections were performed using PEI reagent (Polysciences, 23966) according to the manufacturer's instructions.

## Protein expression and purification

For protein expression in SF9 insect cells (Invitrogen), recombinant baculovirus was generated and amplified according to the manufacturer's protocol. For co-expression of CUL3-His$_6$-RBX1 and His$_6$-KLHL22, suspension SF9 cells were cultured in ESF 921 insect cell culture medium (Sino Biological, MSF1–1L) at 27 °C and infected with recombinant baculovirus encoding (CUL3-His$_6$)-RBX1 and His$_6$-KLHL22. After 72 hrs of incubation, cells were collected by centrifugation at $4000 \times g$ for 15 min. Cell pellets were resuspended in TBS buffer (20 mM Tris-HCl pH 8.0, 140 mM NaCl, 5 mM KCl, 10% glycerol) and cells immediately lysed for protein purification or flash-frozen in liquid nitrogen and stored at −80 °C for later protein purification.

Cell suspensions supplemented with 0.5 mM Tris(2-carboxyethyl) phosphine (TCEP, Hampton, HR2-801) and 1 mM phenylmethanesulfonylfluoride (PMSF, Sigma-Aldrich, P7626-25G) was lysed using an ultrasonic cell crusher (XinChen). After centrifuging twice at $8000 \times g$ at 4 °C for 30 min, the supernatant supplemented with 20 mM imidazole was incubated with Ni-IDA beads (Smart-Lifescience) for 2 h at 4 °C. The bead-protein complexes were washed sequentially with high salt buffer (20 mM Tris-HCl pH 8.0, 500 mM NaCl, 1% glycerol, 0.5 mM TCEP) supplemented with 0 mM, 20 mM, and 40 mM imidazole (Sigma-Aldrich, V90015-500g). The fusion protein was then eluted from the beads with elution buffer (20 mM Tris-HCl pH 8.0, 500 mM NaCl, 1% glycerol, 1 mM TCEP, 300 mM imidazole). The eluted protein was concentrated (Amicon Ultra-15 10KD, Millipore, UFC901096) and further purified by SEC using a Superose 6 increase 10/300 column (GE Healthcare, 29091596) or Superdex 200 increase 10/300 column (GE Healthcare, 28990944) which had been pre-equilibrated with low salt buffer (20 mM Tris-HCl pH 8.0, 150 mM NaCl, 1% glycerol, 1 mM TCEP). The peak fraction was collected and concentrated, and then the purified protein was flash-frozen in liquid nitrogen and stored at −80 °C until needed.

For protein expression and purification in BL21(DE3) *E.coli* cells (TransGen Biotech, CD701-02), plasmids encoding His$_6$-MBP-KLHL22$^{1–178}$, His$_6$-MBP-KLHL12$^{1–161}$, His$_6$-MBP-NRF2$^{1–100}$ were first transformed into BL21(DE3) *E.coli* cells and cultured in Luria-Bertani (LB) medium at 37 °C to an optical density at 600 nm (OD600) of 0.8 before inducing protein expression with 0.1 mM isopropyl-1-thio-β-D-galactopyranoside (IPTG, Inalco, 1758-1400-25g). After 10 hrs of protein expression at 18 °C, cells were collected and protein purified as described above for the insect cell expressed protein, except that the cell lysate was centrifuged only once. The His-MBP tag was removed using TEV protease.

## Cryo-EM sample preparation and data collection

Before cryo-EM sample preparation, the purified CUL3-RBX1-KLHL22 complex was further processed by using an adapted GraFix protocol to exclude protein impurities and to stabilize the complex. A 10–30% glycerol gradient with a 0–0.05% glutaraldehyde gradient (40 mM HEPES pH 7.5, 150 mM NaCl) was prepared and loaded with solution containing CUL3-RBX1-KLHL22 complex. After centrifuging at $110,000 \times g$ at 4 °C for 16 h, the gradient was fractionated and analyzed by SDS-PAGE and proteins stained with Coomassie brilliant blue. Fractions with minimal protein impurities were combined and centrifuged. Buffer exchange (20 mM Tris-HCl pH 8.0, 150 mM NaCl, 1 mM TCEP) was performed to remove glycerol. The sample quality was evaluated using negative-stain EM.

The cryo-EM grids containing CUL3-RBX1-KLHL22 complex were prepared using a Vitrobot Mark IV (Thermo Fisher) operated at 4 °C and 100% humidity. The Quantifoil Au R1.2/1.3 300-mesh grids were glow-discharged using the plasma cleaner. A 3 μl sample of CUL3-RBX1-KLHL22 complex concentrated to 2.0 mg/ml was applied to the discharged grids and blotted for 5 s with filter paper and plunge-frozen into liquid ethane cooled by liquid nitrogen. The grids were stored in liquid nitrogen for data collection.

To determine the structure of CRL3$^{KLHL22}$, a total of 5446 micrograph stacks were acquired on a FEI Titan Krios transmission electron microscope at 300 kV equipped with a K2 summit direct electron detector (Gatan). Serial-EM[54] was used for automated data collection. Micrographs were taken at a magnification of 81,000× and defocus values from −1.5 μm to −2.0 μm, yielding a pixel resolution of 0.4105 Å px$^{-1}$. Each stack was exposed for 4.4 s with an exposure time of 0.11 s per frame, resulting in 40 frames per stack. The total dose was 50 e$^-$ Å$^{-2}$ for each stack.

## Cryo-EM data processing and model building

CryoSPARC[55] was used for cryo-EM data analysis of CRL3$^{KLHL22}$. Movie alignment was performed by MotionCor2[56] and the parameters of the contrast-transfer function (CTF) were estimated on the motion corrected sum of frames using CTFFIND4.1[57]. Micrographs (5033) were selected based on the CTF fit resolution, average defocus, and ice thickness. The 2D classes selected from blob picking using a subset of 100 micrographs were used as templates for template-based picking. A total of 876,192 particles were boxed out. After four consecutive rounds of 2D classification to exclude "bad particles", three ab initio 3D references were reconstructed by using selected particles. After heterogeneous refinement, a subset of 142,416 particles was selected for non-uniform refinement using $C_2$ symmetry and yielded a reconstruction with an overall resolution of 3.77 Å. The resolution of the dimer interface of CRL3$^{KLHL22}$ was improved to 3.3 Å after local refinement. The masks used for local refinement were created using UCSF Chimera[58]. Local refined maps were sharpened using deepEMhancer[59] and combined using Phenix[60]. Model building and refinement were conducted using COOT[61] and Phenix[60].

## Cryo-EM study of CUL3$^{\Delta1-24}$-RBX1-KLHL22$^{1-178}$ complex

For the preparation of the CUL3$^{\Delta1-24}$-RBX1-KLHL22$^{1-178}$ complex, purified KLHL22$^{1-178}$ was incubated with CUL3$^{\Delta1-24}$-RBX1, concentrated and excess unbound CUL3$^{\Delta1-24}$-RBX1 removed by SEC using Superdex 200 increase 10/300 column which had been pre-equilibrated with low salt buffer. The preparation of cryo-EM grids and data collection procedures were the same as for the wild-type CUL3-RBX1-KLHL22 complex. A total of 1418 micrograph stacks were acquired. The workflow for processing the CUL3$^{\Delta1-24}$-RBX1-KLHL22$^{1-178}$ complex is shown in Supplementary Fig. 6. In brief, movie alignment and CTF estimation were performed by MotionCor2[56] and CTFFIND4.1[57], respectively. A total of 137,638 particles were picked and particle cleaning was performed by 2D classification, ab initio reconstruction, and heterogeneous refinement with 3 classes. Then, a total of 74,279 particles from 2D classification were selected for the reconstruction of (CUL3$^{\Delta1-24}$-RBX1)$_1$-(KLHL22$^{1-178}$)$_2$ complex, and the resolution was refined to 3.94 Å. Local refined maps were sharpened using deepEMhancer[59]. A total of 5070 particles from the 2D classification were selected for the reconstruction of the (CUL3$^{\Delta1-24}$-RBX1)$_2$-(KLHL22$^{1-178}$)$_2$ complex, and the resolution was refined to 8.5 Å.

## S-tag pulldown assay

For S-tag pulldown assays, HEK293T cells were transfected with plasmids encoding KLHL22-Stag together with CUL3$^{WT}$-mCherry, CUL3$^{\Delta1-13}$-mCherry, or CUL3$^{\Delta1-24}$-mCherry. After 48 hrs, cells were collected and washed once with cold PBS buffer (137 mM NaCl, 2.7 mM KCl, 10 mM Na$_2$HPO$_4$, 2 mM KH$_2$PO$_4$), and then lysed in 0.5% NP-40 buffer (20 mM Tris-HCl pH 8.0, 140 mM NaCl, 5 mM KCl, 0.5% NP-40, 1 mM PMSF) supplemented with protease inhibitor cocktail (Roche). Protein concentration in the cell lysates was determined using BCA kit (Thermo Scientific). Equal amounts of cell lysates were incubated with S-tag beads at 4 °C for 4 h with slow rotation, and then the beads were washed with buffer (20 mM Tris-HCl pH 8.0, 140 mM NaCl, 5 mM KCl, 0.1% NP-40, 1 mM PMSF). Samples were boiled with 30 µl 2× SDS-PAGE loading buffer (25 mM Tris pH 7.5, 2% (w/v) SDS, 0.1% (w/v) bromophenol blue, 10% (v/v) glycerol, 1% (v/v) β-mercaptoethanol) for 8 min and analyzed by Western Blotting as described below.

## In vivo ubiquitination assay

To measure the E3 enzyme activity of CRL3$^{KLHL22}$(CUL3$^{WT}$), CRL3$^{KLHL22}$(CUL3$^{\Delta1-13}$), and CRL3$^{KLHL22}$(CUL3$^{\Delta1-24}$), HEK 293T cells were transfected with plasmids encoding CUL3$^{WT}$-RBX1, CUL3$^{\Delta1-13}$-RBX1, or CUL3$^{\Delta1-24}$-RBX1, respectively, together with S-tag-KLHL22, DEPDC5-mCherry and His-ubiquitin. Cells were lysed with buffer (6 M guanidinium-HCl, 10 mM Tris, 100 mM sodium phosphate pH 8.0, 5 mM β-mercaptoethanol, 5 mM imidazole), and lysates incubated with Ni-IDA beads for 2 hrs at room temperature with slow rotation. The beads were washed once with cell lysis buffer, once with pH 8.0 wash buffer (8 M urea,10 mM Tris, 100 mM sodium phosphate pH 8.0, 0.1% (v/v) Triton X-100, 5 mM β-mercaptoethanol), three times with pH 6.3 wash buffer (8 M urea,10 mM Tris, 100 mM sodium phosphate pH 6.3, 0.1% (v/v) Triton X-100, 5 mM β-mercaptoethanol), and eluted with elution buffer (200 mM Imidazole, 5% (w/v) SDS, 150 mM Tris-HCl pH 6.7, 30% (v/v) glycerol, 720 mM β-mercaptoethanol and 0.0025% (w/v) bromophenol blue), and then analyzed by Western Blotting as described below.

## In vitro ubiquitination assay

CUL3$^{WT}$-RBX1, CUL3$^{\Delta1-13}$-RBX1, and CUL3$^{\Delta1-24}$-RBX1 were first neddylated for prior to carrying out an in vitro ubiquitination assay. Neddylation was performed by incubating 5 µM CUL3-RBX1 (CUL3$^{WT}$-RBX1, CUL3$^{\Delta1-13}$-RBX1, or CUL3$^{\Delta1-24}$-RBX1) with 6.3 µM NEDD8, 800 nM NAE1/UBA3 (E1), 200 nM UBE2M (E2) in reaction buffer (20 mM HEPES pH 7.5, 150 mM NaCl, 10 mM MgCl$_2$, 2 mM ATP, 1 mM DTT) at 27 °C for 90 min.

For the in vitro ubiquitination assay, 1 µM pre-neddylated CUL3$^{WT}$-RBX1, CUL3$^{\Delta1-13}$-RBX1, or CUL3$^{\Delta1-24}$-RBX1 was incubated with a pre-prepared ubiquitination mix containing 1 µM UBE1, 1 µM UBE2D1, 1 µM KEAP1, 1 µM MBP-NRF2$^{1-100}$, 2 mM ATP, 10 mM MgCl$_2$, 20 mM HEPES pH 7.5 at 4 °C. Reactions were started by incubating the reaction mix at 30 °C for the indicated times and were stopped by adding 1x SDS loading buffer. Ubiquitination efficiency was then analyzed by SDS-PAGE and Western blotting as described below.

## Western blotting

Equal amounts of protein samples were separated on 4–20% gradient polyacrylamide gels and then transferred to polyvinylidene difluoride membranes under constant current conditions, and blots were probed with the indicated primary antibodies overnight at 4 °C. Subsequently, membranes were washed in TBST buffer (25 mM Tris pH 7.5, 140 mM NaCl, 3 mM KCl, 0.1% Tween-20) four times and further probed with secondary antibody (HRP conjugated goat anti-rabbit, Pierce, 31460) (diluted 1:5000). Immunoreactive bands were detected with the western blotting luminol reagent (Santa Cruz Biotechnology Inc., sc-2048). The following primary antibodies were used in this study: anti-CUL3 (Abcam, Ab75851, EPR3196Y, 1:1000), anti-CUL3 (Abclonal, A16455, 1:1000), anti-mCherry (Abcam, Ab213511, EPR20579, 1:1000), anti-KLHL22 (Proteintech, 16214-1-AP, 1:2000), anti-DEPDC5 (Abcam, Ab213181, EPR20497-23, 1:1000), anti-beta-actin (MBL, #PM053, 1:5000), anti-UBA1 (Abcam, ab181225, EPR14204(B), 1:5000), anti-KEAP1 (Abcam, Ab227828, EPR22664-26, 1:1000), anti-NEDD8 (Abcam, Ab81264, Y297, 1:1000), anti-Stag (Abcam, Ab180958, EPR12996, 1:1000), anti-UBE2D1 (Abclonal, A1951, 1:1000) or anti-MBP (Abcam, Ab119994, EPR4744, 1:5000), anti-Flag (Sigma, F3165, M2, 1:2000). All antibodies used are commercially available and extensively used.

## Analytical size-exclusion chromatography

For analytical SEC, Superdex 200 Increase 10/300 GL column (GE Healthcare) was pre-equilibrated with low salt buffer (20 mM Tris-HCl pH8.0, 150 mM NaCl, 1% glycerol, 1 mM TCEP). Purified proteins were mixed at a 1:1 mole (4.9 µmol) ratio of (CUL3$^{WT}$-RBX1): KLHL22$^{1-178}$, (CUL3$^{\Delta1-13}$-RBX1): KLHL22$^{1-178}$, (CUL3$^{\Delta1-24}$-RBX1): KLHL22$^{1-178}$ in a total volume of 200 µl for 10 min on ice. The incubated complexes were then injected into the column. An equivalent amount of KLHL22$^{1-178}$ in a total volume of 200 µl was analyzed alone. Eluted fractions (500 µL) were analyzed by SDS-PAGE and stained with Coomassie brilliant blue. The same procedures were performed for analyzing the molecular sizes of CUL3$^{WT}$-RBX1- KLHL12$^{1-161}$, CUL3$^{\Delta1-13}$-RBX1-KLHL12$^{1-161}$, CUL3$^{\Delta1-24}$-RBX1-KLHL12$^{1-161}$, KLHL12$^{1-161}$, CUL3$^{WT}$-RBX1-KEAP1, CUL3$^{\Delta1-13}$-RBX1-KEAP1, CUL3$^{\Delta1-24}$-RBX1-KEAP1, and KEAP1 alone. The same analysis procedures were also performed by using 25 µM KLHL22$^{1-178}$ incubated with 50 µM or 100 µM CUL3-RBX1; 28 µM KLHL12$^{1-161}$ incubated with 56 µM and 112 µM CUL3-RBX1; and 7 µM KEAP1 incubated with 14 µM and 28 µM CUL3-RBX1.

## Multi-angle light scattering coupled with size-exclusion chromatography (SEC-MALS)

For the SEC-MALS assay, a system containing a WTC-015S5 column, Optilab T-rEX refractive index detector, and a miniDAWN TREOS 3-angle MALS detector (Wyatt Technology) was used. The system was pre-equilibrated with low salt buffer as above and 120 µl protein samples at a concentration of 3 mg/ml (for analysis of KLHL22$^{1-178}$, KLHL12$^{1-161}$) or 2 mg/ml (for analysis of CUL3$^{WT}$-RBX1-KLHL22$^{1-178}$, CUL3$^{\Delta1-24}$-RBX1-KLHL22$^{1-178}$, CUL3$^{WT}$-RBX1-KLHL12$^{1-161}$, and CUL3$^{\Delta1-24}$-RBX1-KLHL12$^{1-161}$) were loaded. Data were analyzed using ASTRA software (Wyatt). In Fig. 4g, h and 5d, e, Supplementary Fig. 5c and 7f, the chromatograms show the relative Rayleigh ratios (right Y-axes, red line), relative dRI ratios (right Y-axes, blue line), and calculated molecular weights (left Y-axes, black line).

## Isothermal titration calorimetry (ITC)

For the ITC assay, an ITC200 instrument (Malvern Instruments) was used. CUL3$^{WT}$-RBX1 (120 μM), CUL3$^{\Delta1-13}$-RBX1 (120 μM), and CUL3$^{\Delta1-24}$-RBX1 (120 μM), respectively, were each placed in the injection syringe and titrated into KLHL22$^{1-178}$ (10 μM) in the sample cell in a volume of 300 μl at 25 °C. Each titration consisted of a total of 20 successive injections with 0.4 μl for the first and 2.0 μl for the remainder. The power of each injection was recorded and plotted as a function of time. Data analysis, including background reduction, baseline correction and evaluation, was carried out with MicroCal software. Data were fitted to a one-site model according to standard procedures. The stoichiometry of binding ($N$) and the equilibrium-association constant ($K_A$) were determined based on the fitted data. The equilibrium-dissociation constant ($K_D$) was derived from $K_A$. The same analysis was repeated using KLHL22$^{1-178}$ (120 μM) titrated into, respectively, CUL3$^{WT}$-RBX1 (12 μM), CUL3$^{\Delta1-13}$-RBX1 (12 μM), and CUL3$^{\Delta1-24}$-RBX1 (12 μM).

## Mass photometry

BSA and thyroglobulin (T9145) were obtained from National Institute of Standards and Technology (USA) and Merck, respectively. Mass photometry experiments utilized a TwoMP mass photometer (Refeyn Ltd, UK) recording movies at 475.2 Hz with exposure times of 2.06 ms, adjusted for optimal camera counts and to prevent saturation. The instrument was calibrated prior to measurements using BSA (66 kDa, 132 kDa) and thyroglobulin (660 kDa). Focus was established by adding a fresh buffer (20 mM Tris-Hcl pH 8.0, 150 mM NaCl, 1% Glycerol, 1 mM TCEP) into an 18 μL well, identifying and locking the focal position with the instrument's autofocus. Each acquisition involved adding 2 μL of protein solution at a concentration of 10 nM (KLHL22$^{1-178}$ at 100 nM; CUL3$^{WT}$-RBX1 at 25 nM). All experiments performed at least three times. Data were analyzed using DiscoverMP software (v2023 R2).

## Evolutionary conservation analysis

CUL3 sequences from different species were aligned using Jalview[62] and Clustal software[63]. The species included in the alignment are *Homo sapiens* (Q13618, human), *Mus musculus* (Q9JLV5, mouse), *Xenopus tropicalis* (A4IHP4, western clawed frog), *Gallus gallus* (A0A8V1A7E5, chicken), *Danio rerio* (AAH65357, zebrafish), and *Drosophila melanogaster* (Q9V475, fruit fly).

## Molecular dynamics simulation analysis

The structure of CUL3$^{1-155}$-KLHL22$^{22-178}$ was used for molecular dynamics simulation. The simulation system was placed in a cubic water box and neutralizing counter ions were added. To mimic physiological conditions, NaCl was subsequently introduced to a final concentration of 150 mM. The protein complexes were centered in the box and placed at least 10 Å from the box edge. Molecular dynamics simulations were carried out by GROMACS 2023, and all simulations are using the CHARMM36 force field. The steepest descent algorithm was used to minimize systems. The system was equilibrated for 100 ps at 310 K and then subjected to molecular dynamics simulation for 1000 ns in the isothermal–isobaric (NPT) ensemble. The short-range electrostatic and Lennard–Jones interactions cutoffs were kept at 12 Å, and long-range electrostatic interactions were calculated using Particle Mesh Ewald methods. The simulation trajectories were analyzed using GROMACS.

## Reporting summary

Further information on research design is available in the Nature Portfolio Reporting Summary linked to this article.

## Data availability

The data supporting this study are accessible from the corresponding authors upon request. The cryo-EM density map of CUL3$^{WT}$-RBX1-KLHL22 and CUL3$^{\Delta1-24}$-RBX1-KLHL22$^{1-178}$ have been deposited in the Electron Microscopy Data Bank under accession number EMD-36961 [https://www.ebi.ac.uk/pdbe/entry/emdb/EMD-36961] and EMD-36987 [https://www.ebi.ac.uk/pdbe/entry/emdb/EMD-36987], respectively. The corresponding atomic coordinates were deposited in the RCSB Protein Data Bank under accession numbers 8K8T and 8K9I. The focused map and consensus map related to EMD-36961 have been deposited in the Electron Microscopy Data Bank under accession number EMD-39719 [https://www.ebi.ac.uk/pdbe/entry/emdb/EMD-39719] and EMD-39720 [https://www.ebi.ac.uk/pdbe/entry/emdb/EMD-39720], respectively. The cryo-EM density map of CUL3$^{WT}$-RBX1-KLHL22 refined with C1 symmetry, which show more clear density corresponding to CUL3 NA motif, have been deposited in the Electron Microscopy Data Bank under accession number EMD-39725 [https://www.ebi.ac.uk/pdbe/entry/emdb/EMD-39725]. The crystal structure of CUL3-KLHL11, CUL3-KEAP1, CUL3-SPOP, and CUL3-A55 is obtained from the RCSB Protein Data Bank under accession numbers 4AP2, 5NLB, 4EOZ and 6I2M, respectively. The cryo-EM density map of CRL3$^{KLHL22}$ determined by Teng et al.[37] was obtained from Electron Microscopy Data Bank under accession number EMD-37247 [https://www.ebi.ac.uk/pdbe/entry/emdb/EMD-37247]. Reagents generated in this study will be made freely available for academic research purposes and/or may require a Materials Transfer Agreement. Source data are provided with this paper.

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

## Acknowledgements

We thank Xuemei Li, Zhenxi Guo, Xia Pei, Guopeng Wang, and Rui Jiao from the Laboratory of Electron Microscopy (Peking University) for providing kind help with cryo-EM sample screening and data collection; Dr. Lei Zhang from Peking University Medical and Health Analysis Center for negative staining EM data collection; Rongrong Dai and Dr. Mo Hu (Mass Spectrometry Core, Changping Laboratory, China) for conducting mass photometry measurements. We thank Dr. Susan Cole from Queen's University for valuable feedback and constructive suggestions on this manuscript. This study is supported by grants to Y.Y. including the National Key Research and Development Program of China (2021YFA1300601, Y.Y.), National Natural Science Foundation of China (key grant 82030081 and 81874235, Y. Y.); Shenzhen High-level Hospital Construction Fund and Shenzhen Basic Research Key Project (JCYJ20220818102811024, Y.Y.). L.L. is supported by the National Natural Science Foundation of China (grants 31800626 and 32171224, L.L.). Y.Y. is the Scholar of the Lam Chung Nin Foundation for Systems Biomedicine. The data processing is supported by the high-performance computing platform of Peking University.

## Author contributions

Y.Y. conceived the project. W.W., L.L. and Z.D. conducted the biochemical experiments and analyzed the results. P.Z. prepared the cryo-EM samples and collected the cryo-EM data. W.W. and L.L. performed data processing, model building and refinement. L.L. and S.Y. performed the molecular dynamics simulation analysis. Y.L., D.D., H.C., H.S., and Y. J. assisted in experiments. Y.M analyzed the cryo-EM data, provided critical ideas for the study, and revised the manuscript. W.W. and Y.Y. wrote the manuscript and revised the manuscript with input from all authors.

## Competing interests

The authors declare no competing interests.
