## [Peer Review File · Nature Communications]

Reviewers' Comments:

Reviewer #1:

Remarks to the Author:

Wang et al. describe the the complex of Cul3-Rbx1 with the substrate adapter KLHL22 using cryo-EM, solution binding, and biochemical activity measurements. Overall, the experiments are well done, although I have concerns about the interpretation and discussion of the results. BTB dimerization and BTB-BACK/Cul3 interface has been extensively described in previous work and established features are sometimes presented as novel discoveries.

Major comments:

1. This MS has significant overlap with the recently published paper by Teng et al. (Structure, published 02 Oct 2023, PMID: 37788672). The Teng paper presents the cryo-EM structure of KLHL22/Cul3-RBX1 alone (8KHP) and in complex with the novel substrate GDH1 (8W4J). The Teng paper does not discuss the Cul3 Nterminus (1-24), but does include descriptions of the conformational changes at the BTB/BACK interface and the dynamics in the Kelch and Cul3 (lines 148-161 and 172-183, Fig. 2, etc. in the current MS).
2. The authors use the term "E3-E3 super-assembly", but this term has been previously used to describe the association of the RBR E3 ligase AriH1 with a CRL E3 assembly (PMID: 33536622). The situation here is very different (CRL3 homodimer), and the use of "E3-E3 super-assembly" here would add confusion to the literature. The term "dimeric CRL3 assembly" has been used in the literature for nearly 20 years to describe the KLHL family of CRL3 complexes and is the more appropriate term for the KLHL22 complex.
3. The description of the core dimer assembly region (lines 163-171, 202-210, Figs. 3, S3, etc.) provides little new insight from previous structures. Essentially all members of the KLHL family have domains swapped N termini and have a hydrophobic interface and it is well established that these form obligate homodimers. The new data on the KLHL22/Cul3 interface is welcome and adds to the literature but needs to be put in its proper context. References 24-27 are included but relevant results from these papers are not always discussed. Two additional papers (PMID: 22632832 and PMID: 23573258) present the BTB/Cul3 interface and describe the phi-x-E motif (KLHL22 residue L83 is "discovered" here (lines 202-210, Figs. 3, S3), but this is the well-known phi residue). Reference 24 describes the Cul3 Nterminal residues with KLHL11, which are also seen in the current MS with KHL22.

4. Role of the Cul3 residues 1-24. This is a major focus of the paper and is also the most problematic.

The Cul3 N terminus (residues 1-24) makes important, but secondary contributions to KLHL/Cul3 binding. This is already established in the literature. The authors overstate the importance of Cul3(1-24) and use the term "essential" at many points of the MS. The Cul3/KLHL interface can be broken down into 3 components in order of decreasing importance: Cul3(H2,H4,H5)/BTB; Cul3(H5)/3Box and Cul3(1-24)/BTB. Thus, the statement at line 189 is not correct. Reference 24 shows that in KLHL11, Cul3(delta1-24) *contributes* to the association (ref 24), but it is not essential. This MS shows similar data with KLHL22: lines 239, 285, etc.: Cul3(delta1-24) binds to KLHL22 with 400 nM affinity – this is still a pretty strong interaction (although not enough for full activity). The motif is not *essential* for binding.

Note that a recent crystal structure of Cul3/KEAP1(KLHL19) (ref 27, PDB ID 5NLB), Cul3(1-24) is disordered (but possibly due to crystal contacts or ligand binding).

Section 226-286, 288-317 Importance of Cul3 Nterm for binding (SEC-MALS, cryo-EM, etc). These sections can be summarized by stating that the Cul3 Nterm deletions bind more weakly. The Cul3 occupancy on the KLHL22 dimer is a function of the Kd and the relative concentrations of KLHL22 and Cul3. Adding more Cul3 would drive more of the complexes to 2:2 complexes. The authors use equimolar concentrations and find that weaker binding Cul3(1-22) leads to more partly occupied states. This is exactly as expected and provides little insight. Why equimolar? How do the concentrations relate to the local concentrations in cells? See PMID: 37236156 for a recent discussion of the regulation and dynamics of CRL assembly (albeit in a monomeric context). Lines 396-402, 320-323, etc.

Line 251-260, 296-299, etc. The change in SEC elution volume should not be described as "smaller complexes", but rather a change in the distribution of species (KLHL22 dimer + one or two Cul3s). Adding more Cul3 would presumably drive the formation of more 2:2 complexes (assuming no co-operativity). The populations of the species will change with Kd and concentration.

There are many statements throughout that confuse this issue. Here is a small selection of the problematic statements:

Line 226-286: section heading: "CUL3 NA motif deletion leads to non-stoichiometric assembly ...").

Line 274: "deletion of the motif leads to the non-stoichiometric assembly"

Line 386: "deletion of the motif lead to the dimeric KLHL22 recruiting only one CUL4-RBX1 subunit".

Etc.

5. A multiple sequence alignment of Cul3 from different species show that while residues 1-24 are fairly conserved, there is significantly less sequence conservation in this region than the rest of the protein.

6. Additional problematic statements (this is just a small sample):

Lines 88-91: "second step of dimeric E3-E3 assembly are not understood", lines 92-96: "questions that remain". I do not agree – these are in the literature and are not outstanding questions.

Line 308: "motif-dependent assembly" (assembly is affected by the Nterm, but is not dependent on it).

Minor comments:

-Extended data Fig 2, 5: images of the representative micrographs is poor

-Video: No labels, unclear.

Reviewer #2:

Remarks to the Author:

A) Summary

The paper by Wang et al. reports the cryo-EM structure of the dimeric Cullin-RING E3 ubiquitin ligase (CRL) 3 with BTB protein KLHL22. The major highlight of the paper is the structural revelation on the conserved N-terminal motif in Cullin 3 (CUL3 NA), which is shown to be critical for E3 dimerization by directly binding to both the KLHL22 subunits. Deletion analyses strongly supported this notion.

Innovation – Role for CUL3 NA in E3 dimerization is novel. It helps better understanding of CRL3 dimerization, which appears to be a common mechanism for the assembly this E3 subclass.

Significance – This paper is a solid contribution to the CRL field by elucidating a critical role for CUL3 NA in CRL3 dimerization. Both the structural and biochemical analyses are largely sound and well supportive of the main conclusion.

Technical quality – The structural and biochemical analyses are highly complementary.

There are quite a few issues on the writing of the paper as detailed below.

B) Overall assessment

This is a largely solid contribution to the ubiquitin field by adding a previously under-appreciated CUL3 NA motif in CRL3 dimerization. I have several comments to improve the manuscript.

C) Critiques

1) Role of Nedd8: Nedd8 is an activator of all canonical E3 CRLs. How Nedd8 conjugation to CUL3 impacts CRL3 dimer is not discussed.

2) Substrate ubiquitination mediated by a CRL3 dimer: It would be helpful for the authors to speculate/model an architecture containing a CRL3 dimer, a bound substrate(s) and a cognate E2(s). The authors mentioned "ubiquitination zone" in the paper. However, without modeling a complex containing E3, substrate and E2, it is difficult to understand what the "ubiquitination zone" means.

3) Fig. 5g: The authors need to show a CUL3 blot that reveals equal amounts of CUL3 wild type and deletion mutants used. The CUL3 wild type and deletion mutants are expected to show difference in gel mobility. This brought about a question on the Nedd8 blot shown. Given the size difference between the wild type and CUL3 deletion mutants (minus 24 and minus 13 amino acids, respectively), how could Nedd8-CUL3 (WT) and Nedd8-CUL3 (deletion mutants) show identical gel mobility?

4) Confusion on the term "licensing:" The authors claim that CUL3 NA licenses CRL3 activation. It is difficult to understand this claim. The CUL3 NA is required for the assembly of dimeric CRL3, as its deletion leads to formation of unbalanced E3 complex, which is much less active in supporting ubiquitination. E3 activation typically refers to mechanisms that help/facilitate the binding of substrate to E3, recruitment of E2 conjugating enzyme, proper orientation of E2~ubiquitin and E3-substrate to align the thiol-ester complex and receptor lysine residue for catalysis, etc. It is unclear to this reviewer why a CUL3 element involved in full E3 complex assembly is called a licensing factor for E3 activation.

5) Methods on in vitro ubiquitination: Concentrations of UBE1 and UBE2D1 should be in micromolar, not millimolar.

6) Page 13: This reviewer is not aware that CUL9 atomic resolution structure is available.

7) Page 15: Typo – “such as CRL4 (ADCAF1) exists in, both tetrameric and dimeric forms.”

REVIEWER COMMENTS

Reviewer #1:

Wang et al. describe the complex of Cul3-Rbx1 with the substrate adapter KLHL22 using cryo-EM, solution binding, and biochemical activity measurements. Overall, the experiments are well done, although I have concerns about the interpretation and discussion of the results. BTB dimerization and BTB-BACK/Cul3 interface has been extensively described in previous work and established features are sometimes presented as novel discoveries.

Major comments:

1. This MS has significant overlap with the recently published paper by Teng et al. (Structure, published 02 Oct 2023, PMID: 37788672). The Teng paper presents the cryo-EM structure of KLHL22/Cul3-RBX1 alone (8KHP) and in complex with the novel substrate GDH1 (8W4J). The Teng paper does not discuss the Cul3 Nterminus (1-24), but does include descriptions of the conformational changes at the BTB/BACK interface and the dynamics in the Kelch and Cul3 (lines 148-161 and 172-183, Fig. 2, etc. in the current MS).

Response: Thank you for drawing attention to the work of other researchers on the E3 ligase CRL3^{KLHL22}. The key findings presented by Teng et al. in their paper can be summarized in two main points: (1) They have successfully identified GDH1 as a novel substrate for CRL3^{KLHL22}, showing that CRL3^{KLHL22} can promote the polyubiquitination of GDH1 *in vitro*. (2) They discovered the dynamic association of three dimeric CRL3^{KLHL22} with a single GDH hexamer.

In our study, we focused on the role of the CUL3 N-terminal motif in regulating the assembly of dimeric CRL3s. While both Teng et al. and our team investigate the same member of CRL3s family, we approached our studies from different angles, shedding light on distinct aspects of CRL3 functionality. We believe that the contributions from both research groups significantly enhance our understanding of CRL3s and collectively advance the field.

We also noticed that Teng et al. very briefly describe the conformational changes at the BTB/BACK interface and dynamics in the Kelch and Cul3 by performing 3DVA analysis in Cryosparc. We are happy to see that several similar discoveries were made by independent research groups. This indeed suggests that the angle between the BTB and BACK domain of KLHL22 different from the KLHL11 was not merely a result of distinct cryo-EM datasets. It is important to highlight that Teng et al. utilized the purified CUL3-RBX1-KLHL22-GDH1 complex from HEK Expi293F cells for Cryo-EM analysis, while we used the purified CUL3-RBX1-KLHL22 complex from SF9 cells. The observed differences between the CRL3^{KLHL22} and CRL3^{KLHL11} are derived from the unique characteristics of CRL3^{KLHL22}.

Furthermore, we propose that dynamics of three Cullin repeats domains (CR1, CR2, and CR3) contribute to the conformational changes of CRL3^{KLHL22}. Given that all Cullin proteins (CUL1, CUL2, CUL3, CUL4A, CUL4B, CUL5, CUL7) contain Cullin repeats domain, it is conceivable that the CR domain may contribute to the dynamics of all other CRLs. This point was never

addressed by Teng et al. or other researchers. Thus, we think that the conformational changes described in our manuscript are still important enough and worth getting the attention of the CRLs fields. As evidence of our commitment to acknowledging relevant contributions, we have incorporated citations to the discoveries made by Teng et al. in our current manuscripts (Lines 187-189 and lines 239-242). Once again, thank you very much for reminding us this issue.

2. The authors use the term “E3-E3 super-assembly”, but this term has been previously used to describe the association of the RBR E3 ligase AriH1 with a CRL E3 assembly (PMID: 33536622). The situation here is very different (CRL3 homodimer), and the use of “E3-E3 super-assembly” here would add confusion to the literature. The term “dimeric CRL3 assembly” has been used in the literature for nearly 20 years to describe the KLHL family of CRL3 complexes and is the more appropriate term for the KLHL22 complex.

Response: Thank you for bringing this to our attention. We sincerely regret using the somewhat awkward term “E3-E3 super-assembly” to describe the dimeric CRL3 assembly. Now, we have removed this phrase and replaced it with the more appropriate term “dimeric CRL3 assembly” to accurately describe the KLHL22 complex in our current manuscript. We appreciate your understanding and diligence in ensuring precise terminology in scientific communication.

3. The description of the core dimer assembly region (lines 163-171, 202-210, Figs. 3, S3, etc.) provides little new insight from previous structures. Essentially all members of the KLHL family have domains swapped N termini and have a hydrophobic interface and it is well established that these form obligate homodimers. The new data on the KLHL22/Cul3 interface is welcome and adds to the literature but needs to be put in its proper context. References 24-27 are included but relevant results from these papers are not always discussed. Two additional papers (PMID: 22632832 and PMID: 23573258) present the BTB/Cul3 interface and describe the phi-x-E motif (KLHL22 residue L83 is “discovered” here (lines 202-210, Figs. 3, S3), but this is the well-known phi residue). Reference 24 describes the Cul3 Nterminal residues with KLHL11, which are also seen in the current MS with KHL22.

Response: Thank you for pointing out these problems. The BTB domain is evolutionarily conserved, although the primary amino acid sequence varies largely across different KLHL family members. It is naturally observed that the BTB domain of KLHL22 takes a similar approach as other BTB domains to form the BTB dimer. It is also natural to speculate that the determined structure of the KLHL22 BTB domain aligns well with other resolved dimeric BTB domains. Now, we have rewritten the description of the KLHL22 BTB dimer and put it in the proper context (Lines 174-177).

Thank you for addressing the matter concerning the “phi-x-E motif”. We apologize for inadvertently overlooking the work by Wesley J. Errington et al., who initially identified the phi-x-E motif within the CUL3 adaptors. In our revised description, we appropriately acknowledge their contribution to this discovery (Lines 218-224). We had intended to remove the sequence alignment results in Supplementary Figure 3c, as a similar analysis has been performed in the work by Wesley J. Errington et al. However, upon closer examination, we observed that their

sequence alignment analysis just covered 23 out of the 42 KLHL proteins, with KLHL22 not included in their list. Recognizing this, we believe that our sequence alignment results encompassing all 42 KLHL members, hold value. They complement the work by Wesley J. Errington et al. and contribute to supporting the broader understanding of the "phi-x-E motif" discovery.

With regard to the N-terminal residues, we have incorporated additional details from the work by Peter Canning et al. (Reference 24 in the previous manuscript). This includes a more comprehensive description of their findings regarding KLHL11 and adds our findings in the proper context (Lines 197-201 and lines 209-210).

4. Role of the Cul3 residues 1-24. This is a major focus of the paper and is also the most problematic.

The Cul3 N terminus (residues 1-24) makes important, but secondary contributions to KLHL/Cul3 binding. This is already established in the literature. The authors overstate the importance of Cul3(1-24) and use the term "essential" at many points of the MS. The Cul3/KLHL interface can be broken down into 3 components in order of decreasing importance: Cul3(H2,H4,H5)/BTB; Cul3(H5)/3Box and Cul3(1-24)/BTB. Thus, the statement at line 189 is not correct. Reference 24 shows that in KLHL11, Cul3(delta1-24) *contributes* to the association (ref 24), but it is not essential. This MS shows similar data with KLHL22: lines 239, 285, etc.: Cul3(delta1-24) binds to KLHL22 with 400 nM affinity – this is still a pretty strong interaction (although not enough for full activity). The motif is not *essential* for binding.

Response: Thank you for bringing this to our attention. We agree that the CUL3/KLHL interface can be broken down into 3 components: Cul3(H2, H4, H5)/BTB, Cul3(H5)/3Box and Cul3(1-24)/BTB. In our original manuscript, our statement was limited to indicating that CUL3/KLHL interface is mainly formed between 3Box and CUL3(H2, H5). This statement did not include all interactions. Now, we have revised this statement to encompass all three components (Lines 79-81, lines 91-96, lines 196-201, and lines 201-205).

We think that all 3 components are important for the affinity between KLHL and CUL3-RBX1 subunit. However, our results showed that the complete CUL3 N-terminus not only provides affinity but plays a definitive role in ensuring the assembly integrity of dimeric CRL3s. Our results show that Cul3(delta1-24) binds to KLHLs with pretty strong interaction, yet only one CUL3 can be recruited to KLHLs, resulting in almost complete loss of E3 ligase activity. Whether the other two components have such an important role still needs further investigation. However, we do believe that the CUL3 N-terminus is important for the assembly integrity and E3 ligase activity of the dimeric CRL3s, even though it may not be deemed essential. To avoid any misunderstanding, we have deleted all such "essential" terms and replaced them with "important" in our revised manuscript. We appreciate your valuable input in addressing this issue.

Note that a recent crystal structure of Cul3/KEAP1(KLHL19) (ref 27, PDB ID 5NLB), Cul3(1-24) is disordered (but possibly due to crystal contacts or ligand binding).

Response: CRL3^{KEAP1} is one of the most important members of CRL3s and has been given much attention during past years. That is also the one reason why we chose CRL3^{KEAP1} as an example to illustrate the important role of CUL3 N-terminus in the assembly of dimeric CRL3^{KEAP1}. Our results show that deletion of the CUL3 N-terminus not only leads to only one CUL3-RBX1 recruited by dimeric KEAP1 but almost leads to loss of interactions between KEAP1 and CUL3-RBX1, which suggests that CUL3 N-terminus plays more important role in the assembly integrity and function of CRL3^{KEAP1}, compared to the CRL3^{KLHL22} and CRL3^{KLHL12}. As matter as fact, we have collected the Cryo-EM data of CRL3^{KEAP1}, and the density map of CRL3^{KEAP1} shows that the CUL3 N-terminus is visible, but the resolution is not as high as CRL3^{KLHL22}. In the crystal structure of CUL3-KEAP1 (PDB ID: 5NLB), the CUL3(1-24) is invisible and we agree with you that possibly due to crystal contacts or ligand binding. It is really interesting and important to further determine how the CUL3 N-terminus regulates the function of CRL3^{KEAP1} under both physiological and pathological conditions.

Section 226-286, 288-317 Importance of Cul3 Nterm for binding (SEC-MALS, cryo-EM, etc). These sections can be summarized by stating that the Cul3 Nterm deletions bind more weakly. The Cul3 occupancy on the KLHL22 dimer is a function of the Kd and the relative concentrations of KLHL22 and Cul3. Adding more Cul3 would drive more of the complexes to 2:2 complexes. The authors use equimolar concentrations and find that weaker binding Cul3(1-22) leads to more partly occupied states. This is exactly as expected and provides little insight. Why equimolar? How do the concentrations relate to the local concentrations in cells? See PMID: 37236156 for a recent discussion of the regulation and dynamics of CRL assembly (albeit in a monomeric context). Lines 396-402, 320-323, etc.

Line 251-260, 296-299, etc. The change in SEC elution volume should not be described as “smaller complexes”, but rather a change in the distribution of species (KLHL22 dimer + one or two Cul3s). Adding more Cul3 would presumably drive the formation of more 2:2 complexes (assuming no co-operativity). The populations of the species will change with Kd and concentration.

There are many statements throughout that confuse this issue. Here is a small selection of the problematic statements:

Line 226-286: section heading: “CUL3 NA motif deletion leads to non-stoichiometric assembly ...”).

Line 274: “deletion of the motif leads to the non-stoichiometric assembly”

Line 386: “deletion of the motif lead to the dimeric KLHL22 recruiting only one CUL4-RBX1 subunit“.

Etc.

Response: Thank you for addressing the concerns regarding our experiment design on the assembly of CUL3-RBX1-KLHL22, CUL3-RBX1-KLHL12, and CUL3-RBX1-KEAP1 complexes. To be honest, we were not considering the absolute local molar concentration of each subunit in the living cell when performing our *in vitro* experiment. In our previous manuscript, we just used equimolar (5 μ M of CUL3^{WT}, CUL3 ^{Δ 1-24}, KLHL22¹⁻¹⁷⁸, KLHL12¹⁻¹⁶¹, KEAP1) to compare the assembled complex with or without the CUL3 N-terminal motif. We think that the μ M level is comparable to the molar concentration of many functional cellular proteins if not considered the highly abundant proteins. Furthermore, the 5 μ M would not use too many purified

proteins and also meet the requirement of the following experiments, such as analytical SEC, SDS-PAGE, SEC-MALS, and Cryo-EM. Your reference to the work by Kheewoong Baek, et al. (PMID: 37236156) mentions that “cellular concentrations of SKP1-Fbps are in substantial excess of CUL1-RBX1”. Given that SCFs and CRL3s all belong to the Cullin-RING E3 ligase family, if both of them take a similar strategy in the cellular environment, it is reasonable to speculate that cellular concentrations of CUL3 adaptors are excess of CUL3-RBX1. In other words, the free CUL3-RBX1 in the cellular environment may be less than the free KLHL22, KLHL12, or KEAP1. However, we acknowledge that these are speculative assumptions, and the true cellular concentrations and ratios remain to be open for further investigation.

We agree that relying solely on equimolar may not be persuasive to get the conclusion that deletion of the CUL3 N-terminal motif results in the recruitment of only one CUL3-RBX1 subunit by CUL3 adaptors. Our speculation is aligned with this concern that adding more CUL3-RBX1 would presumably drive the formation of more 2:2 complexes. In our Cryo-EM data set of CUL3^{Δ1-24}-RBX1-KLHL22¹⁻¹⁷⁸ sample, while the majority of protein particles (94%) are in the state of (CUL3^{Δ1-24}-RBX1)₁-(KLHL22¹⁻¹⁷⁸)₂, we did observe approximately 6% protein particles in the state of (CUL3^{Δ1-24}-RBX1)₂-(KLHL22¹⁻¹⁷⁸)₂. Based on these results, we proposed that CUL3^{Δ1-24}-RBX1-KLHL22¹⁻¹⁷⁸ complex may exist in a state of dynamic equilibrium between the (CUL3^{Δ1-24}-RBX1)₂-(KLHL22¹⁻¹⁷⁸)₂ state and the (CUL3^{Δ1-24}-RBX1)₁-(KLHL22¹⁻¹⁷⁸)₂ state, with the equilibrium favoring the formation of (CUL3^{Δ1-24}-RBX1)₁-(KLHL22¹⁻¹⁷⁸)₂.

To determine whether the CUL3-RBX1 occupancy on the KLHL22 dimer is affected by adding more CUL3-RBX1, we employed two molar ratios (1:2 and 1:4) for SEC analysis. Specifically, 25 μM KLHL22¹⁻¹⁷⁸ was incubated with 50 μM or 100 μM CUL3-RBX1, respectively. The results show that adding more CUL3 does not drive the formation of more (CUL3^{Δ1-24}-RBX1)₂-(KLHL22¹⁻¹⁷⁸)₂ complexes (Response Fig.1 a-d). Similar results were observed with KLHL12¹⁻¹⁶¹ (Response Fig.1 e-h, 28 μM KLHL12¹⁻¹⁶¹ incubated with 56 μM and 112 μM CUL3-RBX1, respectively) and KEAP1 (Response Fig.1 i-i, 7 μM KEAP1 incubated with 14 μM and 28 μM CUL3-RBX1, respectively).

Response Fig. 1 | Adding more CUL3^{Δ1-24}-RBX1 does not drive the formation of more (CUL3^{Δ1-24}-RBX1)₂-KLHLs₂ complexes. | SEC analysis of the effect of the concentration of CUL3-RBX1 on the assembly of the CUL3-RBX1-KLHL22¹⁻¹⁷⁸ (a), CUL3-RBX1-KLHL12¹⁻¹⁶¹ (e), and CUL3-RBX1-KEAP1 (i). (b-d), (f-h), and (j-l) show the SDS-PAGE gel of corresponding SEC fraction of CUL3-RBX1-KLHL22¹⁻¹⁷⁸ (a), CUL3-RBX1-KLHL12¹⁻¹⁶¹ (e), and CUL3-RBX1-KEAP1 (i), respectively.

The precise reason why the deletion of the CUL3 N-terminal motif results in KLHLs only recruits one CUL3-RBX1 subunit requires further investigation. We observed the additional map density, which likely corresponds to the CUL3 aa 1-13, is parallel to the density of the domain-swapping β1' strand of KLHL22 promoter 2 ((Response Fig.2 a and b)). In the previous manuscript, we are not sure if this additional density is noise or belongs to the CUL3 N-terminal. However, this additional density was also existing in the density map (Response Fig.2 c), released on 13 Dec 2023 determined by Teng et al., another independent research group. Our results of molecular dynamic simulations show that the CUL3 N-terminal motif is associated with two KLHL22 subunits. Together with our ITC and SEC results, we now more strongly believe that the additional density corresponds to the CUL3 aa 1-13. We have included relevant results in the current manuscript Figure 3i-k. These results lead us to speculate that CUL3 may engage in multiple inter-subunit interactions, ensuring the recruitment of two CUL3-RBX1 subunits to the dimeric KLHLs. The exact underlying mechanism needs further investigation. In addition, as mentioned earlier, we observed that deletion of the CUL3 N-terminal motif not only leads to only

one CUL3-RBX1 being recruited by dimeric KEAP1, but also causes loss of interactions between KEAP1 and CUL3-RBX1. This suggests that the CUL3 N-terminal motif plays a more important role in the assembly integrity and function of CRL3^{KEAP1} compared to the CRL3^{KLHL22} and CRL3^{KLHL12}.

Response Fig. 2 | CUL3 N-terminal motif interacts with domain-swapping β 1' strand of KLHL22 promoter 2. (a and b), Cryo-EM density map of the CUL3 N-terminal motif (blue) interacts with KLHL22 promoter 2 (orange). Overlay of CUL3 N-terminal motif cartoon model on the density map. (c), Cryo-EM density map (EMD-37247, released on 13 Dec 2023 by Teng, et al.) of the CUL3 N-terminal motif (blue) interacts with KLHL22 promoter 2 (orange).

Based on these results, we think that now using the descriptions of “CUL3 NA motif deletion leads to non-stoichiometric” or “deletion of the motif lead to the dimeric KLHL22 recruiting only one CUL3-RBX1 subunit” are suitable and necessary to describe the distinct role of CUL3 N-terminal motif in the assembly of dimeric CRL3s.

Our previous cryo-EM data showed that CUL3 ^{Δ 1-24}-RBX1-KLHL22¹⁻¹⁷⁸ complex may exist in a state of dynamic equilibrium between the (CUL3 ^{Δ 1-24}-RBX1)₂-(KLHL22¹⁻¹⁷⁸)₂ state and the (CUL3 ^{Δ 1-24}-RBX1)₁-(KLHL22¹⁻¹⁷⁸)₂ state, with the equilibrium favoring the formation of (CUL3 ^{Δ 1-24}-RBX1)₁-(KLHL22¹⁻¹⁷⁸)₂. As the above results show adding more CUL3 does not drive the formation of more (CUL3 ^{Δ 1-24}-RBX1)₂-(KLHL22¹⁻¹⁷⁸)₂ complexes. We propose that the formation of (CUL3 ^{Δ 1-24}-RBX1)₂-(KLHL22¹⁻¹⁷⁸)₂ complex may be due to the (CUL3 ^{Δ 1-24}-RBX1)₁-(KLHL22¹⁻¹⁷⁸)₂ complex transiently associated with another CUL3 ^{Δ 1-24}-RBX1 subunit, but very quakily disassociates.

Above all, we are grateful for the critical and instructive comments by the reviewer, which have guided us in reaching more solid conclusions on whether the CUL3 N-terminal motif merely contributes to affinity or indeed plays a role in the assembly integrity of dimeric CRL3s.

5. A multiple sequence alignment of Cul3 from different species show that while residues 1-24 are fairly conserved, there is significantly less sequence conservation in this region than the rest of the protein.

Response: We agree that compared with other regions of CUL3, the CUL3 N-terminal 1-24 is relatively less conserved. Recognizing the significance of the CUL3 CR domain and C-terminal globular domain in scaffold and E3 ligase activity, it is evident that these domains are relatively

more evolutionarily conserved. Upon closer examination, although the CUL3 N-terminal 1-24 is relatively less conserved, we note that the amino acid residues corresponding to interactions with the KLHL22 in our determined CUL3^{KLHL22} and previous determined CUL3^{NTD}-KLHL11^{BTB-BACK} are highly conserved. Moreover, considering the primary amino acid sequence variability among BTB domain-containing proteins, the surface properties of dimeric BTB domain-containing proteins may vary considerably. We hypothesize that the CUL3 N-terminal motif needs to interact with various of BTB domain-containing proteins, and this relatively less sequence conservation may provide the CUL3 N-terminal motif with great potential to interact with different BTB domain-containing proteins.

6. Additional problematic statements (this is just a small sample):

Lines 88-91: “second step of dimeric E3-E3 assembly are not understood”, lines 92-96: “questions that remain”. I do not agree – these are in the literature and are not outstanding questions.

Line 308: “motif-dependent assembly” (assembly is affected by the Nterm, but is not dependent on it).

Response: Regarding the issue in Lines 88-91 and lines 92-96, we have discussed as above, and admitted that our initial description was not suitable. CUL3-RBX1 recruitment by KLHLs is not mainly through 3-box, but through 3 groups interactions: Cul3(H2,H4,H5)/BTB; Cul3(H5)/3Box and Cul3(1-24)/BTB. We have now revised these descriptions (Lines 90-95).

For the issue of “motif-dependent assembly”, we now recognize that dimeric CUL3s assembly is not only dependent on the CUL3 N-terminal motif, but also dependent on other elements, such as H2, H4, H5 of the CR1 domain of CUL3. The CUL3 N-terminal motif is distinct from other elements in that it not only contributes to affinity but also plays an important role in the assembly integrity of dimeric CUL3s. It is acknowledged that, in the absence of the additional association elements, dimeric CUL3s may still fail to assemble. Now, we have revised this description to more accurately convey the essence as “CUL3 N-terminal motif regulated assembly”.

We have checked throughout our manuscript and revised inappropriate statements.

Minor comments:

-Extended data Fig 2, 5: images of the representative micrographs is poor

Response: We apologize for using these down-sampled representative micrographs; we have now replaced them with high-resolution images and included a scale bar.

-Video: No labels, unclear.

Response: Now, we have added relevant labels and descriptions. Thank you for bringing these issues to our attention.

Reviewer #2 (Remarks to the Author):

A) Summary

The paper by Wang et al. reports the cryo-EM structure of the dimeric Cullin-RING E3 ubiquitin ligase (CRL) 3 with BTB protein KLHL22. The major highlight of the paper is the structural revelation on the conserved N-terminal motif in Cullin 3 (CUL3 NA), which is shown to be critical for E3 dimerization by directly binding to both the KLHL22 subunits. Deletion analyses strongly supported this notion.

Innovation – Role for CUL3 NA in E3 dimerization is novel. It helps better understanding of CRL3 dimerization, which appears to be a common mechanism for the assembly this E3 subclass.

Significance – This paper is a solid contribution to the CRL field by elucidating a critical role for CUL3 NA in CRL3 dimerization. Both the structural and biochemical analyses are largely sound and well supportive of the main conclusion.

Technical quality – The structural and biochemical analyses are highly complementary. There are quite a few issues on the writing of the paper as detailed below.

B) Overall assessment

This is a largely solid contribution to the ubiquitin field by adding a previous under-appreciated CUL3 NA motif in CRL3 dimerization. I have several comments to improve the manuscript.

C) Critiques

1) Role of Nedd8: Nedd8 is an activator of all canonical E3 CRLs. How Nedd8 conjugation to CUL3 impacts CRL3 dimer is not discussed.

Response: Thanks for your suggestions. Given that both neddylation and the CUL3 NA motif regulate the E3 ligase activity of CRL3s, it is reasonable to speculate that neddylation and CUL3 NA motif may cooperate with each other to fine-tune the E3 ligase activity of CRL3s. Neddylation activates the CRLs in several ways, including promoting conformational changes of RBX1 RING domain and WHB domain of Cullin proteins, increasing affinities for ubiquitin-loaded E2s, positioning the ubiquitylation activate site proximal to the substrates. A previous study by Weam I Mohamed et al. showed that neddylation can switch the autoinhibited tetrameric CRL4A^{DCAF1} into an activated dimeric state. This study sheds light on how neddylation can activate the CRLs by changing the oligomer state of CRLs. The basic state of CRL3s is dimeric, but higher oligomer states also exist, such as oligomeric SPOP, pentameric CRL3^{KCTD5}, and tetrameric CRL3^{KBTBD2}. While neddylation may not influence the function of the CUL3 NA motif in the assembly of dimeric CRL3s, it is highly possible that the CUL3 NA motif, either alone or together with

neddylated can regulate the higher oligomeric state of CRL3s. Now, we have included a relative discussion about the effect of neddylation on the dimerization of CRL3s (Lines 456-463).

2) **Substrate ubiquitination mediated by a CRL3 dimer:** It would be helpful for the authors to speculate/model an architecture containing a CRL3 dimer, a bound substrate(s) and a cognate E2(s). The authors mentioned “ubiquitination zone” in the paper. However, without modeling a complex containing E3, substrate and E2, it is difficult to understand what the “ubiquitination zone” means.

Response: Thank you for your suggestions. It is really difficult for others to understand the meaning of “ubiquitination zone”, as the term “ubiquitination zone” was not used frequently. Now, we have added the Nedd8, E2, Ub, substrate, and available Lysine in our model (Response Fig. 2). Thanks again for your suggestions.

Response Fig. 2 | Cartoon model of CRL3^{KLHL22} conformational changes. The dynamic of the CUL3 scaffold subunit leads CRL3^{KLHL22} to transition between a compact conformation and a relaxed conformation, which generates a variable ubiquitination zone.

3) **Fig. 5g:** The authors need to show a CUL3 blot that reveals equal amounts of CUL3 wild type and deletion mutants used. The CUL3 wild type and deletion mutants are expected to show difference in gel mobility. This brought about a question on the Nedd8 blot shown. Given the size difference between the wild type and CUL3 deletion mutants (minus 24 and minus 13 amino acids, respectively), how could Nedd8-CUL3 (WT) and Nedd8-CUL3 (deletion mutants) show identical gel mobility?

Response: Thank you for bringing up this crucial issue. It is essential to include the CUL3 blot in our analysis as we are comparing the E3 ligase activity difference resulting from different versions of CUL3 (WT or deletion mutants). When performing this western blotting experiment, we indeed intended to blot the CUL3 using the anti-CUL3 antibody (Abcam, Ab75851) and tried several times. However, we encountered difficulties as we never detected any signal corresponding to the CUL3 bands. When the same membrane was blotted by the Nedd8 antibody (Abcam, Ab81264), the CUL3~NEDD8 bands (CUL3~NEDD8₁ and CUL3~NEDD8₂) were clearly shown. We suspect that neddylation might be blocking the epitope required for recognition by that anti-CUL3 antibody (Abcam, Ab75851). Now, we have purchased a new CUL3 antibody (Abclonal, A16455) from another company and repeated the CUL3 blotting. We can see that an equal amount of CUL3(WT) and CUL3 (deletion mutants) was used. However, we noticed that the difference in gel mobility may not be significantly evident (Response Fig. 4, lane 4 vs lane 5). This may be

attributed to the use of a 4%-20% SDS-PAGE gel to separate our protein sample, as the CUL3~NEDD8 was about in the 110 kDa position, and at that position, the gel gradient may not efficiently separate CUL3(WT) and CUL3(deletion mutants), due to their subtle 1 kDa difference. In addition, we also included the UBE2D1 blot as shown (Response Fig. 4). The suggestions by the reviewer have been invaluable in refining our results for supporting our conclusion that the complete CUL3 NA motif plays an important role in the E3 ligase activity of CRL3s. Thank you for your guidance.

Response Figure 4 | E3 activity analysis using an *in vitro* ubiquitination assay supply with CUL3 and UBE2D1 blotting.

4) Confusion on the term “licensing:” The authors claim that CUL3 NA licenses CRL3 activation. It is difficult to understand this claim. The CUL3 NA is required for the assembly of dimeric CRL3, as its deletion leads to formation of unbalanced E3 complex, which is much less active in supporting ubiquitination. E3 activation typically refers to mechanisms that help/facilitate the binding of substrate to E3, recruitment of E2 conjugating enzyme, proper orientation of E2~ubiquitin and E3-substrate to align the thiol-ester complex and receptor lysine residue for catalysis, etc. It is unclear to this reviewer why a CUL3 element involved in full E3 complex assembly is called a licensing factor for E3 activation.

Response: We apologize for the confusion caused by using “licensing” to describe the regulatory role of the CUL3 NA motif in CRL3s’ E3 ligase activity. This term might not be entirely suitable, considering that the CUL3 NA motif can directly regulate the assembly of dimeric CRL3s, subsequently exerting an indirect influence on the E3 ligase activity. Right now, we have no evidence to support the CUL3 NA motif participate in the substrate binding, E2 recruitment, or

proper orientation of E2~ubiquitin and E3-substrate, as you mentioned above. Therefore, using the term “licensing” may not be suitable for this condition. Now, we have revised these inappropriate descriptions throughout our manuscript. Thank you for bringing this matter to our attention.

5) Methods on in vitro ubiquitination: Concentrations of UBE1 and UBE2D1 should be in micromolar, not millimolar.

Response: We did not correctly change the font on “m” to Symbol (μ). Sorry for our carelessness. (Lines 620)

6) Page 13: This reviewer is not aware that CUL9 atomic resolution structure is available.

Response: We previously mentioned that CUL9 also contains three CR repeats domains based on the predicted structure by Alphafold2. However, since the atomic resolution structure of CUL9 has not been determined, we have decided to remove this claim (Line 398).

7) Page 15: Typo – “such as CRL4 (ADCAF1) exists in, both tetrameric and dimeric forms.”

Response: Thank you for reminding us of this Typo error, we have now carefully checked our whole manuscripts to correct such mistakes. (Line 454)

Reviewers' Comments:

Reviewer #1:

Remarks to the Author:

The authors have been very responsive to my comments and have made several changes to their MS to address my concerns, including additional data. Overall, the data presented are very good. However, the main concerns from my first review remain and I still have problems with the interpretation of the results.

1a. "Regulation" by Cul3 residues 1-24.

The authors have changed the term "mediates" to "regulates" in the revised MS in response to my earlier review. "Regulation" is a prominent feature of the MS and is presented in the Title, Abstract, etc. There are about 20 instances of "CUL3 NA motif regulates" (or variations of this idea) throughout the MS. This is an unfortunate choice. Just as "mediates" was not an appropriate term, there is no evidence for "regulation" as far as I know: there are no PTMs or Cul3 isoforms that would provide a mechanism for regulation (modulation of activity) via Cul3 N terminus in a natural biological system. The Nterm is simply part of the interaction surface between Cul3 and KLHL22 and has no regulatory function. I expect that other engineered mutations in the PPI interface (Cul3 helix H2, or BTB, or 3Box) could generate the same binding and activity results as the Cul3 N-terminal deletion mutants presented here - mutations to weaken but not abolish Cul3 binding. The Cul3 Nterm simply **contributes** to the binding interaction, and the authors show this with non-physiological truncations of the Nterm. This is not regulation.

1b. "Quality control system"

Line 113-116, 420, etc. There is no evidence that the Cul3 Nterm acts as a quality control mechanism in a natural biological system.

Page 4, Lines 95-100: I do not agree with the statements/questions in this section (these questions set up the rest of the paper). In my opinion, the framing presents false questions. Cul3 binding to KLHL proteins is well understood and supported by a wealth of published data. As to regulation, the authors frame their work as an extension of reference 32, but reference 32 presents a true, biological system that regulates BTB dimer formation in cells.

2. Improper use of "non-stoichiometric"

Lines 107, 250, 298, 967 and many others, including title to Fig 4. This is not the meaning of "non-stoichiometric". Molar ratios of 1:2, 2:2 etc. are all stoichiometric interactions. Non-integral ratios (e.g. 1:1.8) is non-stoichiometric. Individual molecular complexes are necessarily stoichiometric. A population may have an overall non-stoichiometric ratio if it is a mixture of complexes with different stoichiometries (molar ratios). Thus, the complexes

described here are stoichiometric with molar ratios of 2:2 and 1:2. This is a minor issue and can easily be fixed by cleaning up the wording.

3. Interpretation of stoichiometry/binding data

A central idea in the MS is that the stoichiometry of the complex is affected by the Cul3(d1-23) deletion. I expressed my concern about the interpretation of the binding/stoichiometry in my earlier review, and the authors have responded with additional SEC-MALS data (new Fig. 5). This is appreciated, but the new data do not resolve the issue.

From the rebuttal letter: "Based on these results, we proposed that CUL3d1-24-RBX1-KLHL221-178 complex may exist in a state of dynamic equilibrium between the (CUL3d1-24-RBX1)₂-(KLHL221-178)₂ state and the (CUL3d1-24-RBX1)₁-(KLHL221-178)₂ state, with the equilibrium favoring the formation of (CUL3d1-24-RBX1)₁-(KLHL221-178)₂."

I don't think that this is the right way to treat the system. A valid question is whether the two Cul3 binding sites on the KLHL homodimer are independent (i.e. have the same K_d) or whether there is cooperativity between the sites (i.e. binding at one site affects the K_d at the other site). Most KLHL proteins (including Kelch) are 2-fold symmetric and have two non-overlapping binding sites for Cul3. There does not seem to be a conformational change in the KLHL proteins upon Cul3 binding, and so presumably the two sites act independently. i.e. binding at one site does not depend on the status of the other site. If there is no cooperativity, then each of the two Cul3 binding events will have the same K_d and the populations of the 0:2, 1:2 and 2:2 species will depend on the concentration of the KLHL and Cul3 proteins (independent binding).

To argue otherwise (i.e. the truncation of Cul3 "favors" the formation 1:2 complexes) is in fact saying that there is strong negative cooperativity: binding at one site weakens (or in the extreme case, prevents) binding at the other site. This appears to be the model shown in Fig 5i: truncation of Cul3 results in negative cooperativity. This is the final figure that summarizes the MS, and I do not find that this model is reasonable. In my opinion, the arrows in Fig 5i should be from 0:2 to 1:2 to 2:2 (unless the authors are proposing that the two Cul3's need to bind simultaneously to KLHL22 - this would be very difficult to explain since Cul3 is monomeric). If there is strong negative cooperativity, it would be an interesting result. However, the use of SEC and particle populations in cryo-EM, are not very good ways to measure the cooperativity of binding. The ITC data in Fig. 4 b,c,d show that the K_d weakens with truncation of the Cul3 N terminus, but the binding stoichiometry remains about the same, since there is no change of the molar ratio of the midpoint of the curve. This is consistent with non-cooperative binding (i.e. the K_d for each of the two binding sites appears to stay the same).

SEC-MALS is not a good method to measure the K_d and stoichiometry of complexes that are undergoing fast exchange on the timescale of the run. If the off-rate is fast, the complexes can dissociate as they are separated in the column. Since the the CuI3d1-24 truncation results in weaker binding, it is not surprising that 2:2 complexes are difficult to detect by SEC-MALS. ITC is a more quantitative measure for binding affinities and does not appear support the negative cooperativity model.

Reviewer #2:

Remarks to the Author:

The authors have addressed all my previous concerns in the revision.

Point-by-Point Response

Reviewer #1:

The authors have been very responsive to my comments and have made several changes to their MS to address my concerns, including additional data. Overall, the data presented are very good. However, the main concerns from my first review remain and I still have problems with the interpretation of the results.

1a. “Regulation” by Cul3 residues 1-24.

The authors have changed the term “mediates” to “regulates” in the revised MS in response to my earlier review. “Regulation” is a prominent feature of the MS and is presented in the Title, Abstract, etc. There are about 20 instances of “CUL3 NA motif regulates” (or variations of this idea) throughout the MS. This is an unfortunate choice. Just as “mediates” was not an appropriate term, there is no evidence for “regulation” as far as I know: there are no PTMs or Cul3 isoforms that would provide a mechanism for regulation (modulation of activity) via Cul3 N terminus in a natural biological system. The Nterm is simply part of the interaction surface between Cul3 and KLHL22 and has no regulatory function. I expect that other engineered mutations in the PPI interface (Cul3 helix H2, or BTB, or 3Box) could generate the same binding and activity results as the Cul3 N-terminal deletion mutants presented here - mutations to weaken but not abolish Cul3 binding. The Cul3 Nterm simply *contributes* to the binding interaction, and the authors show this with non-physiological truncations of the Nterm. This is not regulation.

Response: Thank you for bringing this matter to our attention. We apologize for using “regulation” for this condition, as no studies have reported the existence of PTM or CUL3 isoforms on CUL3 N-terminal. We have replaced “regulation” with “contributes” throughout our manuscripts. We appreciate your clarification on the precise meaning of “regulation”.

1b. “Quality control system”

Line 113-116, 420, etc. There is no evidence that the Cul3 Nterm acts as a quality control mechanism in a natural biological system.

Page 4, Lines 95-100: I do not agree with the statements/questions in this section (these questions set up the rest of the paper). In my opinion, the framing presents false questions. Cul3 binding to KLHL proteins is well understood and supported by a wealth of published data. As to regulation, the authors frame their work as an extension of reference 32, but reference 32 presents a true, biological system that regulates BTB dimer formation in cells.

Response: In our current manuscripts, our primary focus is on employing *in vitro* biochemical assay to illustrate the role of CUL3 N-terminal in the assembly and E3 ligase activity of CRL3s. While *in vitro* experiments provide a more direct approach, we also realize that to demonstrate the CUL3 N-terminal acts as a quality control mechanism in a natural biological system still need *in vivo* results. Future work will be dedicated to investigating the *in vivo* role of CUL3 N-terminal and exploring its potential involvement in disease. To align with this clarification, we have removed such description of “quality control mechanism” throughout our manuscripts.

Reference 32, along with reference 31, Mena et al. demonstrates a beautiful dimerization quality control mechanism which is essential for BTB homodimer formation in biological system, supported by extensive *in vitro* and *in vivo* experiments. We are impressed by the cell’s evolution of such a precise quality control mechanism to guarantee the BTB homodimer formation. In fact, our work was inspired by these studies. We acknowledge that, compared with the work by Mena et al., describing our work as one part of “dimerization of quality control” may be inappropriate in the current version. Therefore, we have removed Line 95-100 in our revised manuscript.

2. Improper use of “non-stoichiometric”

Lines 107, 250, 298, 967 and many others, including title to Fig 4. This is not the meaning of “non-stoichiometric”. Molar ratios of 1:2, 2:2 etc. are all stoichiometric interactions. Non-integral ratios (e.g. 1:1.8) is non-stoichiometric. Individual molecular complexes are necessarily stoichiometric. A population may have an overall non-stoichiometric ratio if it is a mixture of complexes with different stoichiometries (molar ratios). Thus, the complexes described here are stoichiometric with molar ratios of 2:2 and 1:2. This is a minor issue and can easily be fixed by cleaning up the wording.

Response: Thank you for clarifying the precise meaning of “non-stoichiometric”. We realize that we had misunderstood the meaning of “non-stoichiometric”, and thus we have corrected these inaccuracies throughout our revised manuscripts.

3. Interpretation of stoichiometry/binding data

A central idea in the MS is that the stoichiometry of the complex is affected by the Cul3(d1-23) deletion. I expressed my concern about the interpretation of the binding/stoichiometry in my earlier review, and the authors have responded with additional SEC-MALS data (new Fig. 5). This is appreciated, but the new data do not resolve the issue.

From the rebuttal letter: “Based on these results, we proposed that CUL3d1-24-

RBX1-KLHL221-178 complex may exist in a state of dynamic equilibrium between the (CUL3d1-24-RBX1)₂-(KLHL221-178)₂ state and the (CUL3d1-24-RBX1)₁-(KLHL221-178)₂ state, with the equilibrium favoring the formation of (CUL3d1-24-RBX1)₁-(KLHL221-178)₂.”

I don't think that this is the right way to treat the system. A valid question is whether the two Cul3 binding sites on the KLHL homodimer are independent (i.e. have the same K_d) or whether there is cooperativity between the sites (i.e. binding at one site affects the K_d at the other site). Most KLHL proteins (including Kelch) are 2-fold symmetric and have two non-overlapping binding sites for Cul3. There does not seem to be a conformational change in the KLHL proteins upon Cul3 binding, and so presumably the two sites act independently. i.e. binding at one site does not depend on the status of the other site. If there is no cooperativity, then each of the two Cul3 binding events will have the same K_d and the populations of the 0:2, 1:2 and 2:2 species will depend on the concentration of the KLHL and Cul3 proteins (independent binding).

To argue otherwise (i.e. the truncation of Cul3 “favors” the formation 1:2 complexes) is in fact saying that there is strong negative cooperativity: binding at one site weakens (or in the extreme case, prevents) binding at the other site. This appears to be the model shown in Fig 5i: truncation of Cul3 results in negative cooperativity. This is the final figure that summarizes the MS, and I do not find that this model is reasonable. In my opinion, the arrows in Fig 5i should be from 0:2 to 1:2 to 2:2 (unless the authors are proposing that the two Cul3's need to bind simultaneously to KLHL22 - this would be very difficult to explain since Cul3 is monomeric). If there is strong negative cooperativity, it would be an interesting result. However, the use of SEC and particle populations in cryo-EM, are not very good ways to measure the cooperativity of binding. The ITC data in Fig. 4 b,c,d show that the K_d weakens with truncation of the Cul3 N terminus, but the binding stoichiometry remains about the same, since there is no change of the molar ratio of the midpoint of the curve. This is consistent with non-cooperative binding (i.e. the K_d for each of the two binding sites appears to stay the same).

SEC-MALS is not a good method to measure the K_d and stoichiometry of complexes that are undergoing fast exchange on the timescale of the run. If the off-rate is fast, the complexes can dissociate as they are separated in the column. Since the the Cul3d1-24 truncation results in weaker binding, it is not surprising that 2:2 complexes are difficult to detect by SEC-MALS. ITC is a more quantitative measure for binding affinities and does not appear support the negative cooperativity model.

Response: We acknowledge the importance of investigating whether the recruitment of two CUL3-RBX1 by the BTB dimer is cooperative or independent. Given the absence of an overlapping binding site for CUL3 within the BTB dimer in all resolved BTB-CUL3 structures, we agree that the binding of CUL3 may be independent. Our SEC, SEC-MALS, and Cryo-EM data show that BTB dimer can only recruit only one

CUL3-RBX1 when CUL3 N-terminal is truncated. We also agree that this can be explained by a fast off-rate, especially considering the almost similar molar ratio of the midpoint of the ITC curve. While the results of newly performed mass photometry assay are again consistent with our SEC, SEC-MALS results, showing the loss of dimer formation upon CUL3 N-terminal deletion (Supplementary Fig 5d-i), these results can also be attributed to the decreased binding affinity due to CUL3 N-terminal truncation. Therefore, we refrain from drawing conclusion on whether two CUL3 binding is cooperative or independent at this moment. Future studies may be necessary to further investigate whether two CUL3 binding is cooperative or independent. In addition, regarding the possibility of conformation changes in BTB dimers during CUL3 binding, our cryo-EM structure of KLHL22¹⁻¹⁷⁸₂-(CUL3^{Δ1-24}-RBX1)₁ indicates that the BTB protomer2, which does not bind to CUL3^{Δ1-24}-RBX1, is more dynamic, and we cannot determine its structure at high resolution, hindering us to get a definitive conclusion on BTB dimer undergoes conformational change. Altogether, based on our current results, we agree to interpret our results as the off-rate is fast and have revised our current manuscripts accordingly. We sincerely appreciate your valuable discussions and suggestions, which have significantly contributed to a more comprehensive explanation of our data.

We acknowledge that the model in Fig. 5i in our previous manuscripts may not accurately depict the actual process of recruiting two CUL3-RBX1 by the BTB dimer. In response, we have redesigned the model, introducing the BTB₂-(CUL3-RBX1)₁ intermediate. For the dimeric BTB₂-(CUL3^{Δ1-24}-RBX1)₂, we applied a 50% transparency, reflecting its fast dissociation observed in our SEC, SEC-MALS and cryo-EM data. We hope that the revised cartoon model may be better aligned with the underlying assembly process.

Our exploration into CUL3 N-terminal may just offer preliminary evidence for comprehending assembly of CRL3s family. Apart from the dimeric form, CRL3s can also exist as tetrameric, pentameric, and octameric structure. Further investigation, especially in various physiological and pathological conditions, may still necessary to comprehensively grasp the important role of CUL3 N-terminal. Exploring the exact mechanism of CUL3 N-terminal in the assembly of CRL3^{KEAP1} may be a good choice, as our results show that almost complete lost interaction between KEAP1 and CUL3 with N-terminal truncation. Additionally, the presence of cancer-associated mutations within CUL3 N-terminal may provide another layer of evidence for its potential pathological role.

Many thanks for guiding us in the right direction to fully understand the exactly role of CUL3 N-terminal in the assembly of dimeric CRL3s. We really appreciate your expertise in biochemistry and feel fortunate to have the opportunity, through this review, to delve deeply into the CRL3s assembly process with you.

Reviewers' Comments:

Reviewer #1:

Remarks to the Author:

The authors have addressed all of my concerns in the revised manuscript. The mass photometry results are a nice addition.